# Anthropogenic Land Use Change and Adoption of Climate Smart Agriculture in Sub-Saharan Africa

Sarah Ephrida Tione [1,*] , Dorothy Nampanzira [2], Gloria Nalule [2], Olivier Kashongwe [3,4] and Samson Pilanazo Katengeza [5]

1   Department of Planning, Ministry of Agriculture, Lilongwe P.O. Box 30134, Malawi
2   Department of Livestock and Industrial Resources, College of Veterinary Medicine, Animal Resources and Biosecurity, Makerere University, Kampala P.O. Box 7062, Uganda
3   Department of Engineering for Livestock Management, Leibniz Institute of Agricultural Engineering and Bioeconomy, 14469 Potsdam, Germany
4   Department of Animal Sciences, Faculty of Agriculture, Egerton University, P.O. Box 536, Nakuru 20115, Kenya
5   Directorate of Research and Outreach, Lilongwe University of Agriculture and Natural Resources (LUANAR), Lilongwe P.O. Box 219, Malawi
*   Correspondence: sarahtione@gmail.com; Tel.: +265-999-544-665

**Abstract:** Compelling evidence in Sub-Saharan Africa (SSA) shows that Climate-Smart Agriculture (CSA) has a positive impact on agricultural productivity. However, the uptake of CSA remains low, which is related to anthropogenic, or human-related, decisions about CSA and agricultural land use. This paper assesses households' decisions to allocate agricultural land to CSA technologies across space and over time. We use the state-contingent theory, mixed methods, and mixed data sources. While agricultural land is increasing, forest land is decreasing across countries in SSA. The results show that household decisions to use CSA and the extent of agricultural land allocation to CSA remain low with a negative trend over time in SSA. Owned land and accessing land through rental markets are positively associated with allocating land to CSA technologies, particularly where land pressure is high. Regarding adaptation, experiencing rainfall shocks is significantly associated with anthropogenic land allocation to CSA technologies. The country policy assessment further supports the need to scale up CSA practices for adaptation, food security, and mitigation. Therefore, scaling up CSA in SSA will require that agriculture-related policies promote land tenure security and land markets while promoting climate-smart farming for food security, adaptation, and mitigation.

**Keywords:** climate change adaptation; climate-smart agriculture; land allocation; agricultural policy; Sub-Saharan Africa

## 1. Introduction

In Sub-Saharan Africa (SSA), population growth and urbanization continue to put pressure on land use and land productivity, especially in the agricultural sector [1]. SSA is a region where the livelihoods of many people depend on agricultural activities, making land the major household economic asset. The growing need for land in non-agriculture sectors such as housing and urban infrastructure is exacerbating the small landholding size, mostly in areas close to urban centers [2,3]. The high dependency on agricultural land and continuous cultivation is also putting pressure on land productivity in rural areas. Climate change is further affecting agricultural productivity in SSA, given that agricultural production is mainly rainfed. Observation in [4] showed that only 4 to 6 percent of total cultivated land in SSA is under irrigation. Therefore, the high potential for irrigation farming is yet to be fully exploited in this region. In addition, increasing agricultural productivity has mainly focused on intensification and clearing land. These also contribute to Greenhouse Gas (GHG) emissions from land use and agricultural production decisions [5].

According to the 2007 IPCC Fourth Assessment Report, the terrestrial ecosystem in Africa has significantly changed, evidenced by the reduction in vegetation area [6]. The report further indicates that the primary driver has been anthropogenic (human-related) changes in land use, including land area expansion for crop and livestock production at the cost of vegetation and forest cover. Livestock production alone contributes about 14.5 percent of global GHG emissions and nearly half of the agriculture sector's emissions, due to enteric fermentation and land clearing (i.e., animal digestion, feed production, manure management, and forest cover loss) [6]. Therefore, if SSA is to achieve the Sustainable Development Goals, especially Goal 2, which aims at ending hunger, achieving food security, improving nutrition, and promoting sustainable agriculture, agricultural productivity will have to increase in the face of climate change while reducing GHG emissions.

Realizing this need, agricultural policy interventions in SSA have been promoting Climate-Smart Agriculture (CSA) technologies and practices, especially among smallholder farmers. According to the Food and Agricultural Organization (FAO), "Climate Smart Agriculture is an approach that helps to guide actions needed to transform and reorient agricultural systems to effectively support the development and ensure food security in a changing climate" [7]. Mainly, CSA looks at sustainably increasing agricultural productivity and incomes, adapting and building resilience to climate change, and reducing emissions of greenhouse gas. From the time agricultural policies started advancing the use of CSA technologies, empirical evidence shows positive impacts on agricultural production emanating from land use decisions, particularly among smallholder farmers in SSA [8]. However, the uptake of CSA practices in low-income regions such as SSA is still low, with a scale and scope that is considered unsatisfactory across countries [9]. Observations in [9,10] indicated that it is beneficial if farmers adopt a combination of different CSA technologies and less optimal to focus on a single intervention at the farm or national levels. Thus, the spatial variations in the scale and scope of using CSA, within and across countries, are mainly influenced by the context within which farmers operate to adapt and adopt the CSA technologies.

Despite the empirical evidence on the adoption of CSA technologies and their impact on agricultural productivity, an empirical gap exists in context-specific studies that can assist different stakeholders to prioritize appropriate and timely strategic CSA interventions [10]. Therefore, the objective of this paper is to assess the intertemporal and spatial anthropogenic (human-related) changes in land use associated with CSA household decisions in SSA. Specifically, the paper responds to four key questions. Firstly, what are the intertemporal changes in land allocated to a basket of CSA technologies (explained in the next section) at the household level; (2) what is the effect of the source of farmland (inherited and rented) on the extent of land allocated to CSA, where rented land is associated with the additional cost of using agricultural land; (3) how is the covariate risk from lagged rainfall variations (a proxy for climate change) influencing land allocated to CSA practices and land use changes over time; and (4) how are agriculture-related policy strategies promoting the scaling of CSA practices in SSA?

Our understanding is that CSA can only have significant overall impacts on individuals and the economy if the associated practices are adopted at scale and over time. At the same time, the lasting impacts of CSA, especially among resource-poor households, can also be realized if public and private institutions strengthen policy design and implementation and legal efforts that support the adoption and intensity of using these practices. Thus, this study is relevant for policymakers to understand how CSA is orienting agricultural systems to adapt and build resilience to climate change. To understand the context, the study used data from Malawi, Uganda, and Kenya (Figure 1).

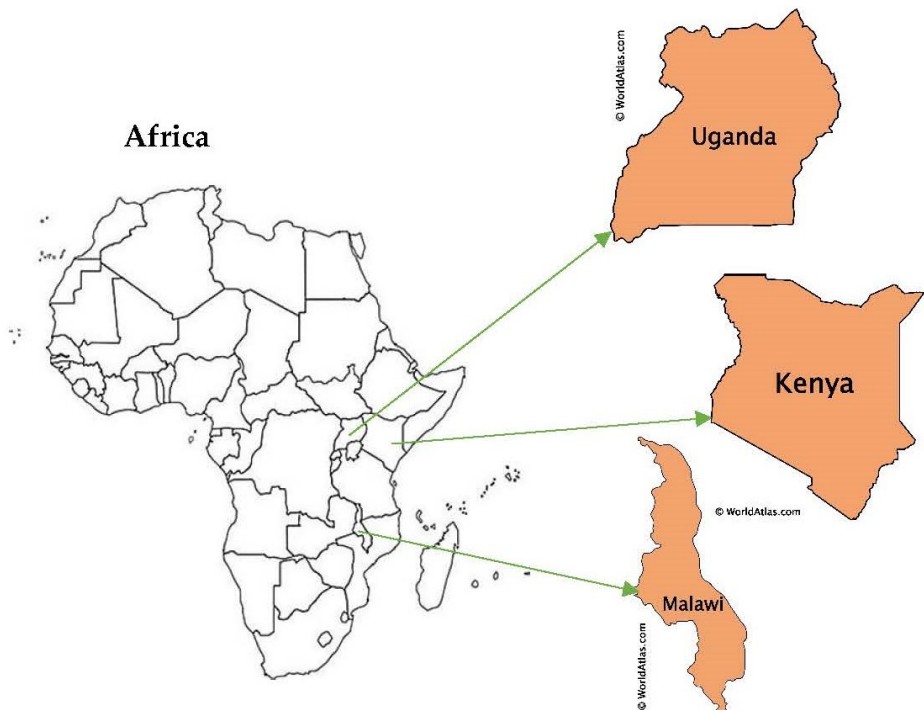

**Figure 1.** Map of Africa showing Kenya, Uganda, and Malawi. Source: https://www.worldatlas.com (accessed on 28 October 2022).

The selection of these countries in SSA is intended to enhance the case-specific empirical studies in this region. Kenya and Uganda are in the Eastern region, while Malawi is in the Southern region of SSA, as shown in Figure 1. The diversity in population, urbanization, land pressure, and use of CSA technologies across these countries is important for context-specific analysis, while the joint review provides a broader analysis of SSA.

## 2. Climate-Smart Agriculture and SSA in Context

Climate-Smart Agriculture (CSA) integrates the economic, social, and environmental dimensions of sustainable development. This implies that there is no "one size fits all" CSA practice recommended universally and also that understanding the context and emphasizing adaptation are important factors for promoting adoption [9]. Policy imperatives for CSA include the need to increase food yields, feed a growing population of nine billion by 2050, mobilize investments for farmers, and reduce GHG emissions. Thus, various technologies and practices have been promoted under CSA, including conservation agriculture that is based on the three farming principles of (1) minimum soil disturbance, (2) organic soil cover, and (3) diversified crop rotations [11]. Overall, CSA includes a variety of technologies and practices, mainly agroforestry, conservation agriculture, crop diversification, adoption of stress-tolerant crop varieties and livestock breeds, and improved water management technologies such as small-scale irrigation [9,12,13]. Specifically, CSA includes the use of organic manure, maize–legume intercropping, and minimum or zero tillage, among other practices [11]. The amount of farmland allocated to one or a combination of CSA technologies and practices determines the use and intensity of adoption among farm households. As land scarcity increases, more land under CSA technologies and practices should improve production and reduce crop area expansion. This could then halt deforestation if complemented with improved natural resources management [14]. This is important, given that agriculture is a key economic activity in SSA.

In Malawi, 80 percent of the economically active population is employed in the agricultural sector. In Uganda, the agricultural sector takes up 68 percent of total employment, while 65 percent of exports in Kenya are agriculture-related [15–17]. The bulk of the population that is engaged in this critical activity is the rural poor, who participate as smallholder

farmers [18]. More than 80% of Uganda's rural population is engaged in agriculture as smallholder farmers [19], with landholdings averaging 2 hectares [20]. In Kenya, agricultural production carried out on smallholder farms averaging 0.2–3 hectares accounts for over 75% of agricultural output [20]. Additionally, smallholder farmers produce the largest percentage of crops consumed domestically, approximately 80% of all food consumed in Malawi [21]. This, therefore, suggests that the adaptation of smallholder farmers to climate change can have profound positive effects on SSA's agricultural sector at large. Moreover, enhancing mitigation contributions from the agriculture sectors alone could enable countries in Eastern and Southern Africa to achieve their ambitious targets to reduce GHG emissions [22].

### 3. Theoretical Framework

Farm households make production decisions under uncertain climatic conditions, or the state of nature. Households allocate factors of production before the state of nature is known. However, acquired experience in climatic conditions and soil fertility on their farms shapes their subjective assessment of production risk in each production season. A farm household whose objective is to maximize production utility or minimize production cost, therefore, makes state-contingent decisions. According to [23], a state-contingent production function assumes that a farm household is faced with $y$ distinct outputs from allocating $x$ distinct inputs with a probability $\pi_s$ of $S$, the state of nature, occurring. Under different states of nature, the farmers' input choice does not determine the output, but different allocations lead to different amounts for costs and output. Thus, with climate risk and subjective assessment of soil fertility, farmers' input choice should minimize production costs under different states of nature.

If we specify a production function as $y = q(A, K)$, where $A$ is land used for production and $K$ is all other inputs (including labour), the cost function can be specified as $Min\ C(r, y) = C[r, q(A, K)]$. The $r$ is the price of inputs, including land rentals. If we assume two states of nature, with probability $\pi_l$ of low outcome and $\pi_h$ of high outcome, the optimal cost function $y^*$ is specified in Equation (1).

$$y^* = min[C\{r, q(A, K)\} : \pi_l y_l + \pi_h y_h = \underline{y} \tag{1}$$

This implies that the household allocation mix of land and other inputs should be risk-substituting or risk-complementing under state-contingent decisions to minimize cost. Therefore, with CSA technologies, farmers would be motivated to choose one or more technologies that are risk-substituting or -complementing at a minimal cost. This implies that a combination of these technologies can be adopted in one parcel of land or different parcels of land owned by the household. For instance, a farmer can combine soil control measures such as terraces or vetiver grass with organic manure and minimum tillage on one piece of land or use different technologies in different areas [12], which can be consolidated to obtain total household land under CSA.

Several factors, including household, community, and governance factors, influence household decisions, for instance, land allocated to CSA [24]. A review of the literature shows that household decisions are a function of household characteristics (gender, size, location, access to community services, education, and experience, and household endowments such as land and asset wealth) [24–26]. Additionally, climatic factors such as rainfall patterns and governance factors such as policies and legislation can influence production decisions over time [24–27]. If a household experiences a deviation in rainfall pattern such as drought and flood in one year, their production decisions are likely to change in the subsequent year to minimize loss or risk [13,23].

Figure 2 conceptualizes the discussed household decision to allocate land for CSA technologies and the factors likely to affect that decision, as discussed. With this theoretical framework and in line with our objectives, we hypothesize three statements:

**H1:** Increase in owned land increases land allocation to CSA technologies.

**H2:** Rented land decreases household land allocation to CSA technologies.

**H3:** Experiencing a decrease in one-year rainfall amount increases land allocation to CSA technologies in the subsequent year.

For hypotheses H1 and H2, the assumption is that households prefer to invest in owned land that is mainly inherited land or land acquired over time through sales markets, which hence is a fixed variable that does not change very often. On the contrary, the user right that comes with rented land can limit CSA investments, particularly for long-term investments [16,28]. The assumption of H3 is that households would respond to immediate rainfall shocks by investing in CSA technologies as a response to climate risk [13].

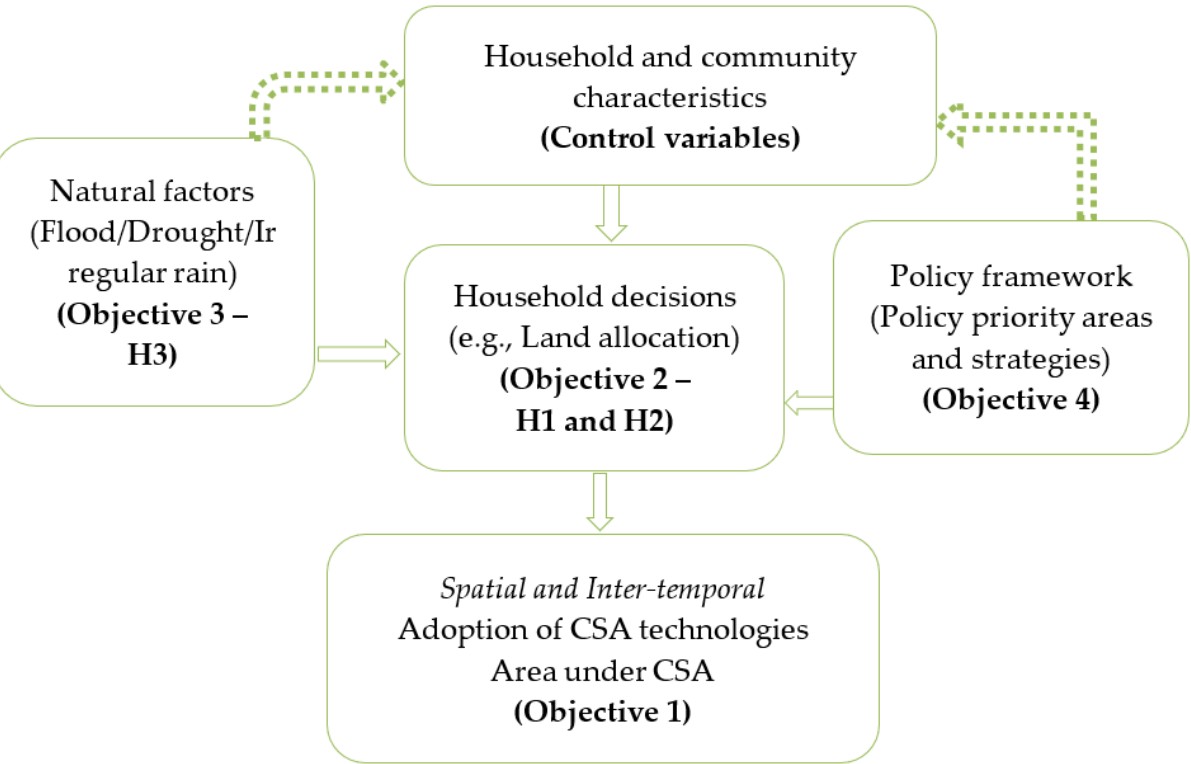

**Figure 2.** Flow diagram of key research variables and objectives. Note: Solid line arrows show a direct pathway in a year, while the dashed arrows show long-term pathways over time.

## 4. Materials and Methods

### 4.1. Data Sources

Two sources of data are used for this study: Living Standards Measurement Surveys (LSMS) and FAOSTAT. LSMS data are periodically collected by the World Bank in developing countries. The panel data collect household and community characteristics, in addition to agricultural production data on land and the use of CSA technologies. The data also provide annual rainfall variables at the community level, which would be important for assessing the covariate risk related to changes in rainfall distribution. For this study, we use the recent three rounds and two rounds of the LSMS data collected in Malawi and Uganda, respectively. The three rounds of data in Malawi were collected in 2013, 2016/17, and 2019/20. In Uganda, the two rounds of LSMS data used were collected in 2014/15, and 2019/20. These data are used to respond to objectives 1 to 3. FAOSTAT data report country-level indicators over time. We use the FAOSTAT data from 1961 to 2018 to assess country-level indicators, more specifically to respond to objective 1 for Kenya, where we could not access the nationwide household survey data. For objectives 2 and 3, based on the available data in Uganda and Malawi, this study considered the household land allocated

to CSA practices. Specifically, we focused on the use of organic manure (livestock manure), minimum tillage, small-scale irrigation, and soil control measures such as terraces.

For objective 4, we conducted a systematic literature review to identify factors and the attributes for effective scaling of CSA through policy analysis in line with the FAO climate-smart agriculture resource book, a benchmark for CSA policies and practices [29]. A Likert scale was used to assess these factors by assigning a score of 0 for the absence of an attribute or a factor for effective CSA scaling; a score of 1 to indicate that an attribute of a factor for effective CSA scaling is present but not elaborated upon; and a score of 2 to indicate that an attribute of a factor for effective CSA is present and elaborate. We then generated a weighted score for each of the factors for effective scaling of CSA. Additionally, the paper used secondary data obtained from policy documents related to climate-smart agriculture for adaptation or mitigation of climate change or to increase food security under this objective. Country-level documents included national agricultural policies and/or national adaptation plans and climate-smart agriculture policies in the three countries. We summarise the country data and data sources in Table 1.

**Table 1.** Data and Data Source.

| Country | Data Type | Period | Source | Objective |
| --- | --- | --- | --- | --- |
| Malawi | FAOSTAT | 1961–2008 | FAO https://www.fao.org/statistics/en/, accessed on 5 October 2021 | 1 |
| | LSMS | 2013, 2016/17, 2019/20 | Word Bank—LSMS https://www.worldbank.org/en/programs/lsms, accessed on 5 October 2021 | 2 & 3 |
| | Secondary data | Over time | Literature review | 4 |
| Uganda | FAOSTAT | 1961–2008 | FAO https://www.fao.org/statistics/en/, accessed on 5 October 2021 | 1 |
| | LSMS | 2014/15, 2019/20 | Word Bank—LSMS https://www.worldbank.org/en/programs/lsms, accessed on 5 October 2021 | 2 & 3 |
| | Secondary data | Over time | Literature review | 4 |
| Kenya | FAOSTAT | 1961–2008 | FAO https://www.fao.org/statistics/en/, accessed on 5 October 2021 | 1 |
| | Secondary data | Over time | Literature review | 4 |

### 4.2. Empirical Strategy

The study used mixed methods (qualitative and quantitative methods) to respond to the four objectives as indicated below.

Objective 1: For this objective, we used non-parametric methods to graphically show the trends in land allocation among farm households using the unbalanced LSMS data across three panel rounds in Uganda and Malawi. In Kenya, we used the FAOSTAT data to show the trends in land use change. The assessment also included the trends of national agricultural land area under irrigation farming across the three countries and household area under different CSA technologies in Malawi and Uganda. Apart from using the FAOSTAT data, to further assess the trends in Kenya, we reviewed the available literature and graphically presented the data on the use of CSAs [30,31].

Objective 2: To respond to this objective, we used parametric methods. Specifically, we used panel data probit and Tobit models for the use and extent of land allocated to CSA. The key explanatory variables in the model were dummies for renting-in land and the size of owned household land. The rent-in dummy is for the household decision to have additional land with user rights for the short to medium term. Owned land sources include inherited and purchased land, for which the household has control rights [32]. Considering that renting-in the land is also a decision variable, the model will control for the household, garden, and community variables in the analysis. This analysis tests H1 and H2 presented above.

Objective 3: Assessing this objective also used the panel probit and Tobit models, with the amount of land allocated to CSA as the dependent variable and variations in one-year-lagged rainfall amount as the key explanatory variable. Each round of LSMS data provides annual rainfall data in the survey period and two previous seasons at the community level or enumeration area to show the covariate risk based on geo-referenced data sources [33]. For example, the recent three rounds of the LSMS data from Malawi have annual rainfall data spanning 6 years. Therefore, we used the average rainfall data across the current and previous survey years to generate the one-year-lagged rainfall deviations for each survey period at the enumeration area.

This variable indicates covariate risk, considering that climate risks affect several households at the same time. With data limitations, the yearly rainfall data were not available in the Uganda LSMS, and the data had limited information on GPS coordinates that could be used to merge other sources of rainfall data with this household LSMS data. Thus, to assess the climatic shock in Uganda, we used a dummy for experiencing irregular rains, drought, and floods at the household level. Considering that households use subjective experience to make state-contingent decisions, the use of this variable is a good proxy for assessing the covariate climatic risks that households face. In line with the theoretical framework, the analysis controlled for household, community, and garden characteristics to account for possible bias in the results without claiming full causality. This analysis assessed H3 presented above.

For both objectives 2 and 3 and to respond to hypotheses H1 to H3, we used the reduced form of the probit and Tobit models specified in Equation (2) [34]. In the equation, $A_{jt}$ is the total land area allocated to a basket of CSA technologies for household $j$ and at time $t$. The variable $A^i$ is rent-in dummy and R is one-year-lagged rainfall amount or dummy for experienced climate shock. As mentioned above, the model controls for farm and household characteristics are given as $X_{jt}$ with $\tau$ for time dummies and $\mu_j + \varepsilon_{jt}$ as the additive error term.

$$A_{jt} = \alpha + DA^i_{jt} + \gamma R_{t-1} + \gamma X_{jt} + \tau + \mu_j + \varepsilon_{jt} \tag{2}$$

Based on the available data, this parametric analysis focused on Malawi and Uganda. It was challenging to obtain nationwide household data that specify CSA household land allocation in Kenya; hence, this analysis does not include Kenya. Although LSMS data were available in Malawi and Uganda, the two countries promote different CSA technologies. To perform a comparative analysis, the dependent variable captures total household land under different CSA technologies. In general, the analysis focused on the following CSA technologies: (1) soil erosion control variables such as terraces, control bunds (stones, earth, or sandbags/gibbons), tree belts, water harvesting bunds, and drainage ditches, (2) use of organic manure, (3) irrigation farming by diverting streams, hand and treadle pumps, motor, or gravity feeding, and (4) land preparation techniques that include box ridges, zero tillage, pit planting, ripping, and minimum tillage. Specifically, the Malawi data recorded these technologies, while the data for Uganda categorized land preparation as either burning or not. Additionally, in Uganda, less than 1 percent of the farm households reported the area under irrigation; hence, the analysis for Uganda computed household land allocated to soil control measures, organic manure, and no burning in land preparation.

Regarding the rainfall shock variable, the model for Malawi used a one-year-lagged rainfall variable, while for Uganda, we used a farmer response dummy on experiencing irregular rain, drought, or flood. Furthermore, the analysis in Malawi tested the upside and downside deviation of one-year rainfall amount from the 6-year mean rainfall amount. With three panel rounds that captured rainfall data at the enumeration area, we used rainfall amounts recorded in 2012, 2013, 2015, 2016, 2018, and 2019 to obtain a 6-year mean. Considering that the survey years are 2013, 2016, and 2019, the upside and downside one-year-lagged rainfall deviations calculate the difference between the amount recorded in 2012, 2015, and 2018 from the overall mean. If the recorded amount was above the mean, that is an upside deviation, and if lower than the mean, then it is a downside deviation. For

ease of interpreting the results, the downside deviations were recorded as positive numbers by multiplying them with a negative one.

For the control variables, the analysis includes sex, age, and education of household head; household size; the share of male labour to total household labour; Total Livestock Units (TLU) per total household labour ratio; one-year-lagged TLU per total household labour ratio; distance to urban centres or urban/rural dummy, household-size-to-worker ratio, and capital asset index. The share of male labour, total livestock units, and capital asset capture household endowment are important for making land decisions and use of CSA technologies. To calculate the total labour for the household, we used the adult equivalent labour in man-days; hence, more adult males should imply more household labour. The household-to-worker ratio captures the effort available for agricultural activities compared to the consumption need at the household level.

To estimate these models, we constructed a balanced panel from the available data in Malawi and Uganda [34]. For Malawi, the total sample across the years was 1990, 2508, and 3178 households in the 2013, 2016, and 2019 panel years, respectively. From this sample, we computed a three-year balanced panel of 1439 households. For Uganda, the sample size was 3165, 3174, and 3077 households in the 2015, 2019, and 2020 panel years, respectively. Focusing on CSA practices, the number of households who responded to the CSA-related question in 2019 was low; hence, it was challenging to obtain representative balanced data across the three survey rounds. The study, therefore, used the data from 2019 and 2020, from which we computed a balanced panel of 407 households. The small sample was due to missing responses on CSA and not households dropping out of the survey, which could be a survey problem and not an enumerator data collection problem. To further enhance the cross-country comparative analysis, these results were discussed in comparison with the results of similar studies in Kenya.

Objective 4: To respond to this objective, we used qualitative descriptive analysis to evaluate the CSA policies present in the country policy documents against the set benchmark. We counted the number of policies targeting climate-smart crop and livestock production and water use in the three countries to come up with the total score of policies related to the CSA practices benchmarked by FAO [29]. Finally, we computed a weighted score based on the number of practices included in each country's policy vis a vis the total number enumerated in the FAO sourcebook.

This assessment allowed a comparison of the inclusion of recommended CSA practices in the national policy documents. Hence, country-level policies were assessed as adhering to, modifying, or not adhering to CSA policies as outlined in the FAO document. Adhering meant policies were adopted as stated in the sourcebook, modifying meant some additional features were added or removed from the definition of the sourcebook without changing the overall meaning, while not adhering meant that the policy was not found in the country documents.

## 5. Results

### 5.1. Descriptive Statistics

This section presents the summary statistics across the three years of panel data for Malawi and two years of panel data for Uganda used in assessing objectives 2 and 3. We also present FAOSTAT data to understand land use patterns in Malawi, Uganda, and Kenya in Figure 3. Table 2 shows that the average household landholding size is 0.5 ha in Malawi and 0.8 ha in Uganda, and households reported to have rented land are 10 and 23 percent in Malawi and Uganda, respectively. This reflects FAOSTAT data in Figure 3, which shows that the agricultural land area in Uganda is twice the size of Malawi (11,962,000 ha for Uganda and 5,738,000 ha for Malawi). Across the three study countries, Figure 3 further shows that the proportion of agricultural land to total land is higher in Uganda, followed by Malawi and Kenya. Interestingly, the proportion of forest area is higher in Malawi, followed by Uganda and Kenya. This trend is consistent over the past 30 years, as forest land has been decreasing in all three countries.

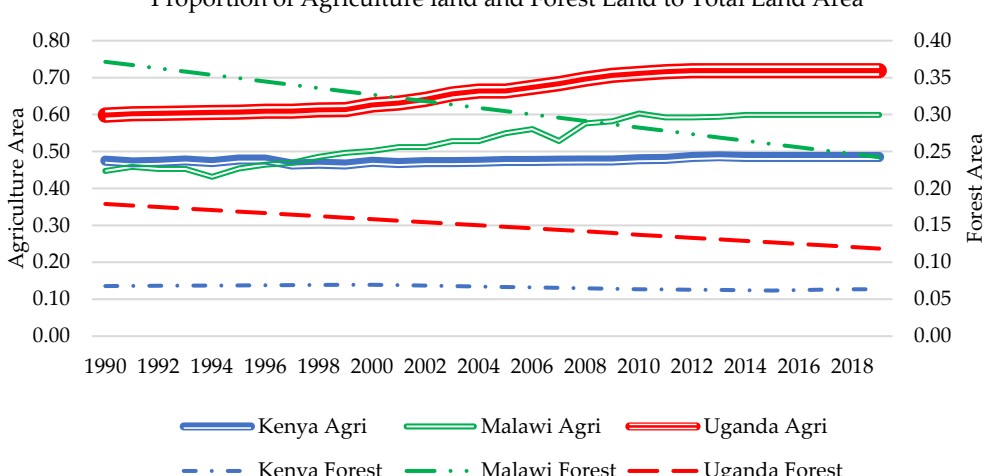

**Figure 3.** Proportion of agriculture land and forest land to total land area in Malawi, Kenya, and Uganda. Source: FAOSTAT data (1990–2018), accessed on 5 October 2021.

**Table 2.** Descriptive Statistics.

| Variable | Unit | Malawi | Uganda |
|---|---|---|---|
| Key variables | | | |
| Area under CSA | Mean (ha) | 0.29 | 0.069 |
| CSA use dummy (1 = Yes) | Percent | 49.2 | 10.69 |
| Rent-in dummy (1 = Yes) | Percent | 10.33 | 23.21 |
| Owned land (GPS measured—ha) | Mean | 0.498 | 0.766 |
| One-year-lag rainfall (per 100 mm) | Mean | 8.70 | |
| One-year-lag upside rainfall deviation (dm) | Mean | 5.35 | |
| One-year-lag downside rainfall deviation (positive dm) | Mean | 4.82 | |
| Irregular rains (1 = Yes) | Percent | | 14.25 |
| Drought (1 = Yes) | Percent | | 48.16 |
| Floods (1 = Yes) | Percent | | 9.58 |
| Control Variables | | | |
| Sex of household head (1 = Female) | Percent | 24.96 | 29.98 |
| Age of household head | Number | 43 | 50 |
| Education of household head | Number | 7 | 7 |
| Share of male labour total labour | Number | 0.41 | 0.43 |
| Household size | Number | 5.26 | 5.5 |
| TLU per labour ratio | Number | 0.10 | 0.61 |
| One-year-lag TLU per labour ratio | Number | 0.10 | 0.42 |
| Distance to urban centre | Km | 24.96 | |
| Reside (1 = Rural) | Percent | | 91.76 |
| Household size per labour ratio | Number | 1.79 | 1.50 |
| Capital asset index | Number | −0.01 | −0.02 |
| Number of observations | Number | | |

Regarding household land area under CSA technology, Malawi has a mean household land allocation of 0.3 ha, while in Uganda, the average land area under CSA is 0.1 ha per household. According to CSA country profiles produced by the organization Climate Change, Agriculture, and Food Security (CCAFS), Malawi mainly promotes conservation agriculture (mulching, minimum or zero tillage), integrated soil fertility management practices (agroforestry, incorporation of organic matter such as mulch, compost, crop residue, and green manure), and use of inorganic fertilizer through the input subsidy program.

In contrast, the profile for Uganda indicates more emphasis on the management of livestock manure (biogas production) and the use of CSA in the production of perennial crops such as intercropping of coffee with banana and legumes with other crops. This could

be the reason why the observed area under CSA is relatively lower in Uganda compared to Malawi [16,17].

The descriptive statistics in Table 2 show that in Malawi, the mean one-year-lagged rainfall amount is 870 mm (mm), while based on downside and upside rainfall variations, the mean deviation is 5.35 decimetres for upside and 4.82 decimetres for downside deviations. This shows that the experienced reduction in rainfall amount was on average 53 mm and the increase was on average 48 mm across the years. For Uganda, we note that 14 percent of the sampled households experienced irregular rains, while 48 percent and 10 percent experienced drought or floods, respectively.

Regarding household characteristics, both Uganda and Malawi have at least 25 percent female-headed households. The mean age of household heads is slightly lower in Malawi compared to the mean age of 50 years for household heads in Uganda. The average education level for household heads, the average share of male labour to total household labour, and household size have no observable differences in the two countries. Uganda data show higher household livestock ownership compared to the households in Malawi, which concurs with the country profiles on CSA for Malawi and Uganda [16,17].

*5.2. Empirical Results and Discussion*

Objective 1: Assessment of intertemporal changes in the proportion of land allocated to CSA technologies

We observe that area under irrigation has been increasing in Malawi and Kenya, but little has changed in Uganda over time (Figure 4). Despite observable changes in the area equipped with irrigation across the three countries, the literature indicates that the percentage of irrigated land out of total agricultural land remains between 4 to 6 percent in these countries [1]. The low area under irrigation particularly under small-scale irrigation, therefore, calls for more policy action if irrigation farming is to promote adaptation to climate change.

**Figure 4.** Land Area equipped with Irrigation. Source: FAOSTAT data (1961–2018), (accessed on 5 October 2021).

Apart from promoting irrigation farming, the intertemporal changes in the household land area allocated to a basket of CSA technologies also show observable changes presented in Figures 5–8. Figures 5 and 6 show the total household area allocated to a basket of CSA technologies in Malawi. Figures 7 and 8 present the household area that indicates different CSA technologies and practices in Uganda. According to Figures 5 and 6, on average, 46 percent of the sampled households allocated land to CSA practices between the years 2013 and 2019. The mean area allocated was 0.27 ha across the years. In Uganda, the trend of households using CSA shows a big drop in the 2019 panel year, where it was only at 4 percent compared to 14 and 13 percent in the 2015 and 2020 survey rounds. Overall, Figure 6 shows that land allocated to CSA has been low in Uganda, averaging at 10 percent, compared to Malawi, which has an average of 46 percent, as shown in Figure 5.

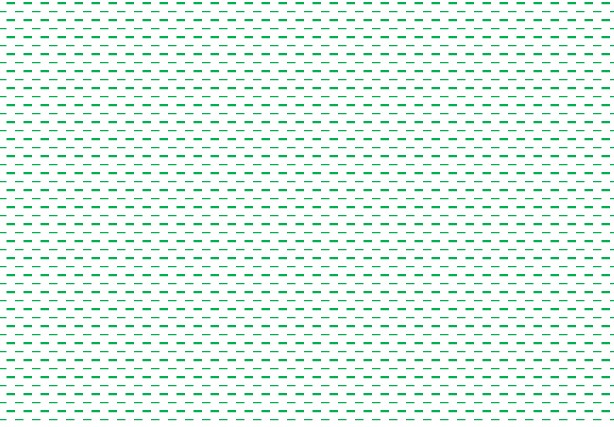

**Figure 5.** Mean Household Land Area (ha) under CSA in Malawi. Source: Computed using balanced LSMS data, (accessed on 5 October 2021).

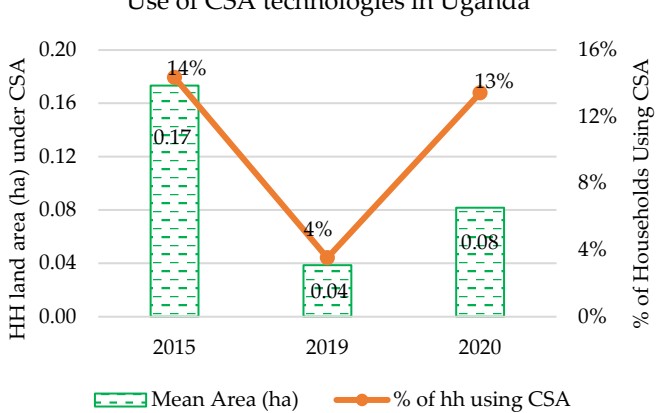

**Figure 6.** Mean Household Land Area (ha) under CSA in Uganda. Source: Computed using balanced LSMS data, accessed on 5 October 2021.

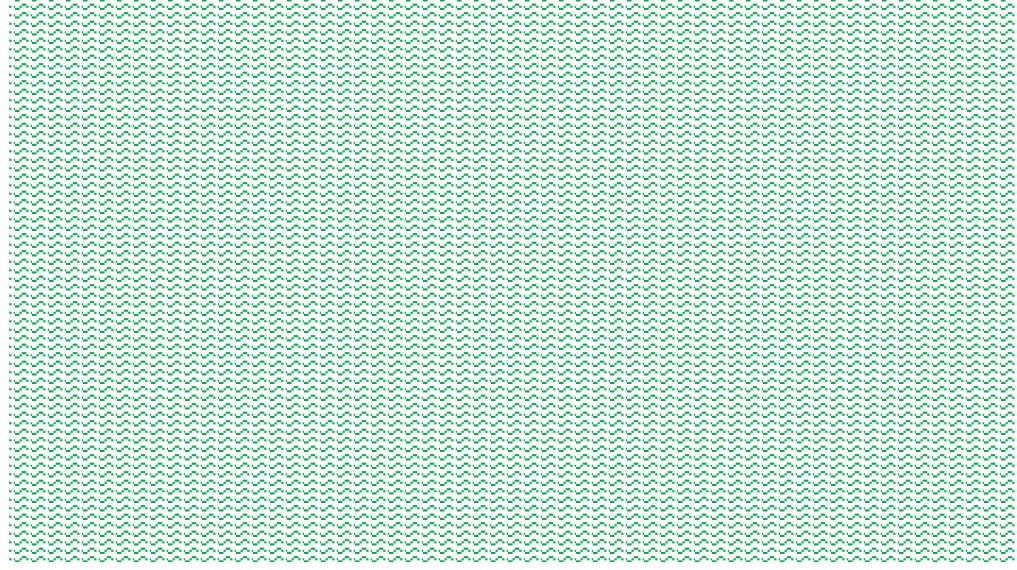

**Figure 7.** Average household land (in acres) allocated to different CSA technologies in Malawi. Source: Computed using balanced LSMS data, accessed on 5 October 2021.

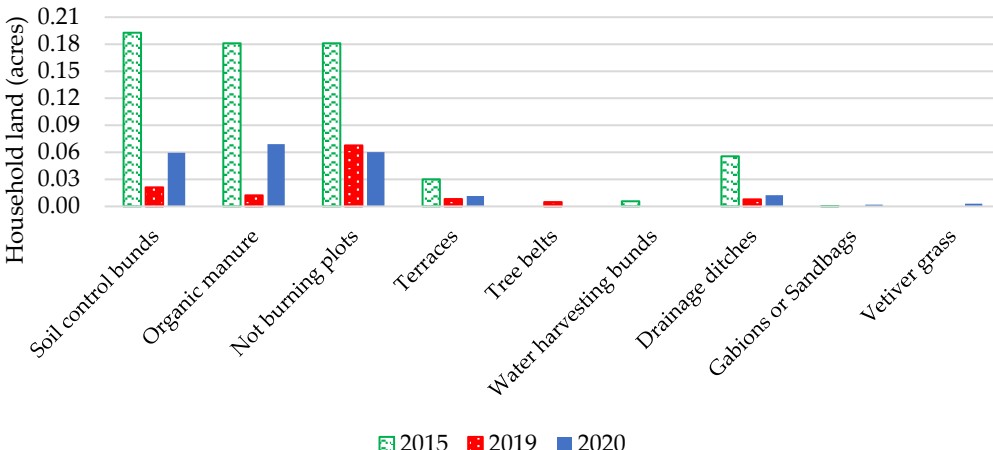

**Figure 8.** Average household land (in acres) allocated to different CSA technologies in Uganda. Source: Computed using balanced LSMS data, (accessed on 5 October 2021).

Compared to Malawi, where the main crop is maize, in Uganda, integrated soil fertility management, agroforestry, crop diversification, and conservation agriculture (crop rotation, mulching, use of green cover crops, and minimum tillage) practices are mainly practiced in the production of tea, banana, rice, coffee, and cassava, among other crops [17]. The use of CSA on perennial crops could imply limited intertemporal land changes in Uganda compared to Malawi or that other related factors affected data collection on land allocated to CSA across the survey rounds. Furthermore, Uganda heavily promotes manure management through the production of biogas as opposed to using manure for farming, which is the reason why the land allocation is small in this country [17].

In general, the trend analysis in this study shows fluctuating trends in households allocating land to a basket of CSA practices and the amount of land under CSA, with an overall negative trend. Such a negative trend can be counter-productive in SSA, knowing that agricultural policies and several projects in these countries are investing in CSA technologies to promote food security and adaptation to climate change [16,17]. Such fluctuating trends are shown further in Figures 7 and 8. From Malawi, the results indicate that the most used technologies are soil control bunds, vetiver grass, and organic manure, and the least used technologies relate to small-scale irrigation, as shown in Figure 7.

In Uganda, farmers are commonly not burning their crop residues in the field, which could be related to mulching (Figure 8). Farmers are also using organic manure and soil control bunds. The data show very low use of irrigation technologies by farmers in Uganda. Considering the short assessment period in this paper, further analysis using long-term data will be important to understand long-term trends in household decisions to allocate land to a basket of CSA technologies, particularly in Uganda. Based on the reviewed literature, Figure 9 shows that in Kenya, the most used CSA technologies are crop management practices (improved crop varieties, legume crop rotation, cover crops, changing planting dates, efficient use of inorganic fertilizers) and field/soil management practices (terraces, planting trees on crop land, rainwater harvesting, farmyard manure, and irrigation) [31,35]. Despite the data challenge across the countries, we now focus our analysis on understanding the factors affecting household land allocation decisions as indicated in objectives 2 and 3.

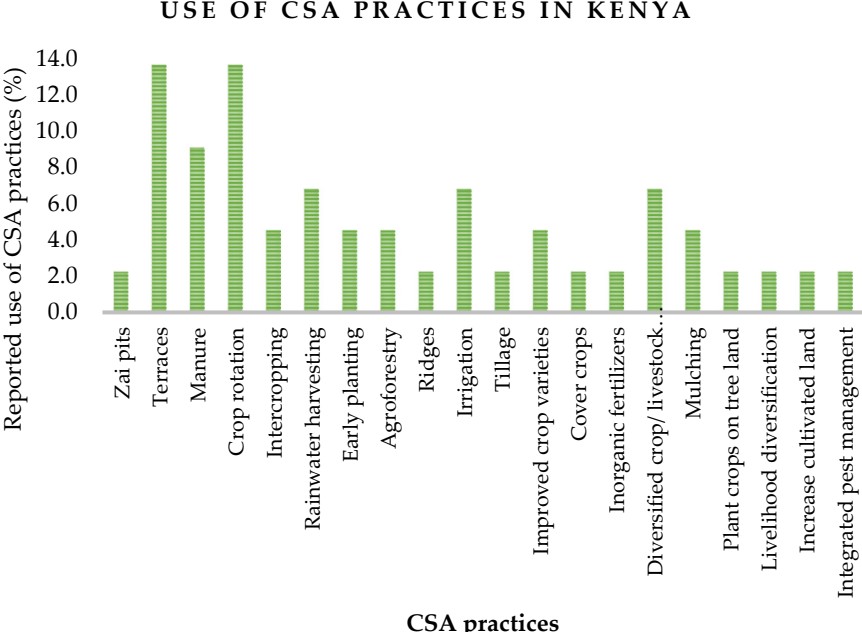

**Figure 9.** Reported percentage use of CSA practices in Kenya. Source: Reviewed literature from Kenya presented in this paper, (accessed on 21 March 2022).

Objectives 2 and 3: Effects of farm household land source and covariate risk from lagged rainfall variations on the extent of land allocation to CSA practices and land use change over time

Tables 3 and 4 present the estimated average margins from the panel probit and Tobit models. In the tables, models 1 and 2 are parsimonious (with only key variables of interest), while models 3 and 4 include the control variables in a stepwise process. In Appendix A, we present the estimated coefficients for these margins in Tables A1 and A2. Under each hypothesis, we discuss the country results and then provide a comparative discussion.

**Table 3.** Estimated margins from Probit and Tobit Models for CSA dummy and Total area allocated to CSA in Malawi.

| Variable | Estimated Margins in Malawi | | | | | | | |
|---|---|---|---|---|---|---|---|---|
| | **Parsimonious Models** | | **Models with Control Variables** | | **Parsimonious Models** | | **Models with Control Variables** | |
| | **Probit1** | **Probit2** | **Probit3** | **Probit4** | **Tobit1** | **Tobit2** | **Tobit3** | **Tobit4** |
| Key Variables | | | | | | | | |
| Rent-in dummy (1 = Yes) | 0.19 **** | 0.19 **** | 0.21 **** | 0.22 **** | 0.10 **** | 0.10 **** | 0.12 **** | 0.12 **** |
| | (0.03) | (0.03) | (0.03) | (0.03) | (0.02) | (0.02) | (0.01) | (0.01) |
| One-year-lag rainfall (per 100 mm) | −0.01 *** | | −0.01 *** | | −0.01 *** | | −0.01 **** | |
| | (0.01) | | (0.00) | | (0.00) | | (0.00) | |
| One-year-lag upside rainfall deviation (dm) | | 0.00 | | 0.00 ** | | −0.00 | | 0.00 |
| | | (0.00) | | (0.00) | | (0.00) | | (0.00) |
| One-year-lag downside rainfall deviation (dm) | | 0.00 | | 0.00 **** | | −0.00 | | 0.00 |
| | | (0.00) | | (0.00) | | (0.00) | | (0.00) |
| Owned land (GPS measured—ha) | | | 0.21 **** | 0.21 **** | | | 0.11 **** | 0.11 **** |
| | | | (0.02) | (0.02) | | | (0.00) | (0.00) |
| Control Variables | | | | | | | | |
| Sex of household head (1 = Female) | | | −0.02 | −0.03 | | | −0.03 ** | −0.03 ** |
| | | | (0.02) | (0.02) | | | (0.01) | (0.01) |
| Age of household head | | | 0.00 | 0.00 | | | 0.00 **** | 0.00 **** |
| | | | (0.00) | (0.00) | | | (0.00) | (0.00) |
| Education of household head | | | −0.00 * | −0.00 * | | | −0.00 ** | −0.00 ** |
| | | | (0.00) | (0.00) | | | (0.00) | (0.00) |
| Share of male labour out of total labour | | | −0.09 ** | −0.09 ** | | | −0.03 | −0.03 |
| | | | (0.04) | (0.04) | | | (0.02) | (0.02) |
| Household size | | | 0.04 * | 0.04 * | | | 0.02 **** | 0.02 **** |
| | | | (0.02) | (0.02) | | | (0.01) | (0.01) |
| TLU per Labour ratio | | | 0.02 | 0.02 | | | 0.00 | 0.00 |
| | | | (0.01) | (0.01) | | | (0.00) | (0.00) |
| One-year-lag TLU per labour ratio | | | 0.00 **** | 0.00 **** | | | 0.00 **** | 0.00 **** |
| | | | (0.00) | (0.00) | | | (0.00) | (0.00) |
| Distance to urban centre | | | 0.00 | 0.00 | | | 0.01 **** | 0.01 **** |
| | | | (0.00) | (0.00) | | | (0.00) | (0.00) |

**Table 3.** *Cont.*

| Variable | Parsimonious Models | | Models with Control Variables | | Parsimonious Models | | Models with Control Variables | |
| --- | --- | --- | --- | --- | --- | --- | --- | --- |
| | **Probit1** | **Probit2** | **Probit3** | **Probit4** | **Tobit1** | **Tobit2** | **Tobit3** | **Tobit4** |
| | | | **Estimated Margins in Malawi** | | | | | |
| Household size per labour ratio | | | −0.01 | −0.01 | | | −0.02 *** | −0.01 *** |
| | | | (0.01) | (0.01) | | | (0.01) | (0.01) |
| Capital asset index | | | −0.08 **** | −0.08 **** | | | −0.04 **** | −0.04 **** |
| | | | (0.01) | (0.01) | | | (0.01) | (0.01) |
| Base Year (2013) | | | | | | | | |
| 2016 year | 0.10 **** | 0.07 **** | 0.09 **** | 0.07 **** | 0.05 **** | 0.03 *** | 0.04 **** | 0.03 ** |
| | (0.02) | (0.02) | (0.02) | (0.02) | (0.01) | (0.01) | (0.01) | (0.01) |
| 2019 year | 0.05 *** | 0.04 *** | 0.07 **** | 0.07 **** | 0.00 | 0.00 | 0.03 *** | 0.03 ** |
| | (0.02) | (0.02) | (0.02) | (0.02) | (0.01) | (0.01) | (0.01) | (0.01) |
| Panel households | 1439 | 1439 | 1439 | 1439 | 1439 | 1439 | 1439 | 1439 |
| Left censored (_n) | | | | | 2205 | 2205 | 2205 | 2205 |
| Uncensored (_n) | | | | | 2085 | 2085 | 2085 | 2085 |
| Observations | 4317 | 4317 | 4317 | 4317 | 4317 | 4317 | 4317 | 4317 |

Standard errors in parentheses = "* $p < 0.1$, ** $p < 0.05$, *** $p < 0.01$, **** $p < 0.001$".

**Table 4.** Estimated margins from Probit and Tobit Models for CSA dummy and Total area allocated to CSA in Uganda.

| Variables | Parsimonious Models | | Models with Control Variables | | Parsimonious Models | | Models with Control Variables | |
| --- | --- | --- | --- | --- | --- | --- | --- | --- |
| | **Probit1** | **Probit2** | **Probit3** | **Probit4** | **Tobit1** | **Tobit2** | **Tobit3** | **Tobit4** |
| | | | **Estimated Margins in Uganda** | | | | | |
| Key Variables | | | | | | | | |
| Rent-in dummy (1 = Yes) | −0.01 | −0.01 | −0.01 | −0.01 | −0.01 | −0.01 | −0.02 | −0.02 |
| | (0.02) | (0.02) | (0.02) | (0.02) | (0.03) | (0.03) | (0.03) | (0.03) |
| Irregular rains (1 = Yes) | 0.00 | | 0.00 | | −0.00 | | −0.00 | |
| | (0.03) | | (0.03) | | (0.03) | | (0.03) | |
| Drought (1 = Yes) | | −0.02 | | −0.01 | | −0.02 | | −0.02 |
| | | (0.02) | | (0.02) | | (0.03) | | (0.03) |
| Floods (1 = Yes) | | 0.02 | | 0.02 | | 0.01 | | 0.02 |
| | | (0.03) | | (0.03) | | (0.04) | | (0.04) |

**Table 4.** *Cont.*

| Variables | Parsimonious Models | | Models with Control Variables | | Parsimonious Models | | Models with Control Variables | |
|---|---|---|---|---|---|---|---|---|
| | **Probit1** | **Probit2** | **Probit3** | **Probit4** | **Tobit1** | **Tobit2** | **Tobit3** | **Tobit4** |
| Owned land (self-reported ha) | | | 0.00 | 0.00 | | | 0.01 | 0.01 |
| | | | (0.00) | (0.00) | | | (0.01) | (0.01) |
| Control Variables | | | | | | | | |
| Sex of household head (1 = Female) | | | 0.00 | 0.00 | | | −0.00 | −0.00 |
| | | | (0.03) | (0.03) | | | (0.03) | (0.03) |
| Age of household head | | | 0.00 | −0.00 | | | −0.00 | −0.00 |
| | | | (0.00) | (0.00) | | | (0.00) | (0.00) |
| Education of household head | | | −0.00 | −0.00 | | | −0.00 | −0.00 |
| | | | (0.00) | (0.00) | | | (0.00) | (0.00) |
| Share of male labour out of total labour | | | −0.05 | −0.05 | | | −0.05 | −0.05 |
| | | | (0.05) | (0.05) | | | (0.07) | (0.07) |
| Total livestock unit to labour ratio | | | −0.04 ** | −0.04 ** | | | −0.05 ** | −0.05 ** |
| | | | (0.02) | (0.02) | | | (0.02) | (0.02) |
| One-year-lag TLU to labour ratio | | | 0.03 | 0.03 | | | 0.03 | 0.03 |
| | | | (0.02) | (0.02) | | | (0.03) | (0.03) |
| Urban (1 = Rural) | | | 0.10 ** | 0.10 ** | | | 0.12 ** | 0.12 ** |
| | | | (0.05) | (0.05) | | | (0.06) | (0.06) |
| Household-to-labour ratio | | | −0.01 | −0.01 | | | −0.01 | −0.01 |
| | | | (0.01) | (0.01) | | | (0.03) | (0.03) |
| Capital asset index | | | 0.04 *** | 0.04 *** | | | 0.05 *** | 0.05 *** |
| | | | (0.01) | (0.01) | | | (0.02) | (0.02) |
| Base Year (2019) | | | | | | | | |
| 2020 panel year | 0.16 **** | 0.16 **** | 0.18 **** | 0.18 **** | 0.21 **** | 0.21 **** | 0.23 **** | 0.22 **** |
| | (0.02) | (0.02) | (0.03) | (0.03) | (0.03) | (0.03) | (0.03) | (0.03) |
| Panel households | 407 | 407 | 407 | 407 | 407 | 407 | 407 | 407 |
| Left censored (_n) | | | | | 727 | 727 | 727 | 727 |
| Uncensored (_n) | | | | | 87 | 87 | 87 | 87 |
| Observations | 814 | 814 | 814 | 814 | 814 | 814 | 814 | 814 |

Standard errors in parentheses = "** $p < 0.05$, *** $p < 0.01$, **** $p < 0.001$".

*The header "Estimated Margins in Uganda" spans all model columns.*

Hypothesis one (H1) stated that an increase in owned land increases land allocation to CSA technologies. The results for Malawi presented in Table 3 concur with this hypothesis, which implies that owning more farmland can encourage the use of CSA technologies at the household level. According to the probit model (probit 4), an increase in owned land increases land allocated to CSA technologies by 21 percentage points. Regarding the extent of land allocation (Tobit 4), a one-hectare increase in owned land increases the household land allocated to CSA technologies by 0.11 ha, which is significant at a 1 percent level. This concurs with the study by [28] in Malawi, which observed that households are more likely to use CSA practices on owned land compared to rented land. From Table 2, we observe that land ownership has no significant effect on land allocated to CSA in Uganda. This could be because in Uganda, the CSA profile shows that CSA-related practices mostly focus on manure management and biogas production compared to using the manure on the farm [17]. The observed insignificant effect in Uganda compared to the significant effect in Malawi, therefore, implies that we can partly fail to reject H1 in SSA. This takes us to the next hypothesis.

Hypothesis two (H2) stated that rented land decreases household land allocation to CSA technologies. The results indicate a significant effect of the rent-in dummy on land allocated to CSA technologies. However, the sign is positive for Malawi in Table 2. This implies that renting-in land can also increase the amount of land allocated to CSA at the household level in Malawi. From the probit model (probit 4), renting-in more land is associated with increased allocation of land to a basket of CSA technologies by 22 percent while increasing the extent by 0.12 ha, as indicated in the Tobit model (Tobit 4), which is significant at 1 percent in Table 3.

Literature shows that landholding size continues to reduce with high land pressure and limited capacity to expand agricultural land to virgin land in Malawi [2,36]. With land pressure, land markets (sales and rental) are developing in Malawi as the main pathway through which households are accessing agricultural land [3,37]. This implies that renting-in land is one of the ways that households are accessing agricultural land and hence the land that households are also allocating to CSA technologies. Using matched tenant and landlord data in Malawi [28] showed that soil management investments by households were higher on owner-operated land compared to rented land. However, the focus of the analysis in the [28] study was not clear on households that mainly depend on rented land, which is a growing pattern in Malawi. Thus, our results build on these studies to show that households that are likely to rent-in more land are also likely to allocate that land for CSA technologies as land pressure continues. In line with the first hypothesis, which showed that owning more land is associated with high likelihood of allocating more land to CSA technologies, renting-in land partly increases one's farmland, hence increasing the likelihood of allocating more land to CSA technologies.

The results for Uganda presented in Table 4 show that renting-in land is not significantly associated with households allocating land to CSA technologies. Interestingly, the sign for this variable shows that renting-in land would decrease land allocated to CSA technologies as expected. Following the argument from the results for Malawi, investing in CSA technologies and practices on rented land could be related to growing land pressure, which can be considered not to be the current situation in Uganda. The trend observed in Kenya by [30] shows an increase in land size by 32% through purchase or leasing for farmers adopting CSA practices. Furthermore, [31] reported an increase in land size by 1 acre (0.4 ha) increased the probability of using a basket of CSA practices by 0.13 to 2.7%. This could be related to the trend observed in Malawi, where population pressure is high. Such trends can be counter-productive if agricultural land becomes scarce. If SSA is to promote CSA, agricultural policies will have to seriously consider access to agricultural land at the household level and promote investing on rented land, in addition to focusing on the adoption and adaptation of CSA technologies. On the adaptation question, we now assess hypothesis three.

Hypothesis H3 stated that a decrease in one-year rainfall amount increases land allocation to CSA technologies. The probit and Tobit model results in Tables 3 and 4 used one-year-lagged rainfall amounts and rainfall deviations in Malawi and households experienced with climate shocks in Uganda. In Table 3, the probit and Tobit model 3 show the effect of one-year-lagged rainfall amount, while probit and Tobit model 4 shows the effect of one-year-lagged upside and downside rainfall deviations in Malawi. In Table 4, the probit and Tobit model 3 show the effect of experiencing irregular rains, while probit and Tobit model 4 show the effect of experiencing drought or floods.

The results for Malawi indicate that an increase in one-year-lagged rainfall amount is associated with a decrease in land allocated to CSA in the subsequent year. From Table 3, if one-year-lagged rainfall increases by 100 mm, households are likely to reduce the land allocated to CSA by 1 percentage point, as shown in probit 3 results, while reducing the area under CSA by 0.01 ha, as assessed in Tobit 3 model results. Intuitively, this implies that a decrease in one-year-lagged rainfall amount should increase land allocated to CSA technologies. Assessing the upside and downside deviation variables, probit model 3 presents a positive and significant effect of both downside rainfall deviation (more associated with drought) and upside rainfall deviation (mores associated with floods) on land allocated to CSA technologies. The observed magnitude effect is too small for economic impact, as evidenced by there being no significant effect in the Tobit model.

These rainfall results imply that households that experience climate shocks are likely to allocate land to CSA technologies in the subsequent year but not likely to increase the extent of land allocation in Malawi. In Uganda, the results show no significant effect, but the observed signs are negative for drought and positive for floods. With the observed effects, we partly reject hypothesis H4 and conclude that observed effects can be context-specific, as explained further below.

The CSA profile for Uganda indicates that 25 percent of cropped land is under root crops, while 17 percent is under banana production, cereals (maize, sorghum, millet, rice) take up 32 percent, and livestock is key in the country [17]. This diversity in crop production and the focus on perennial crops and livestock production in Uganda is different from Malawi, where almost 80 percent of the land is under annual maize production. This could be linked to the results presented in Table 4, where we observe a significant effect of owning livestock in Uganda but not in Malawi. Using the total livestock units to household labour ratio, we note that more livestock compared to household labour is associated with reduced agricultural land area allocated to CSA technologies, as expected. The understanding is that an increase in livestock should increase the availability of manure. However, manure application can be labour-intensive; hence, households with low labour are constrained in using manure in the field compared to using the manure for other uses such as producing biogas if the system is established. Therefore, future analysis can focus on climate-smart technologies under livestock manure management (biogas production) as an adaptation measure and changes in land allocated to climate-smart technologies under perennial crops. Apart from assessing the household context, the study also assessed the country policy context and scaling up of CSA technologies, as specified in objective 4 and as discussed below.

Objective 4: Institutional and policy strategies to promote the scaling of CSA practices

Table 5 summarizes the CSA technologies or practices and shows the possible impact on food security (FS), adaptation (AD), and mitigation (MI), in line with the FAO climate-smart agriculture resource book [29]. Based on this categorization, Tables 6–8 summarize the CSA-related objectives of agriculture policy and National Adaptation Plans in Uganda, Kenya, and Malawi. From the tables, the numbers in brackets with slashes represent the order number of CSA practices by sector (integers), management objectives within the sectors (one decimal place), and practices within management objectives (two decimal places). Figure 10 further summarizes the CSA objectives and shows the overall expected impact areas of CSA policies in the three countries.

**Table 5.** Climate-smart agriculture practices and their potential highest impact on food security (blue), adaptation (green), and mitigation (orange).

| Sector | Management Objective | Practices | Highest Impact (FS / AD / MI) | | |
|---|---|---|---|---|---|
| Crop [1] | Improved crop varieties [1.1] | Conventional breeding (e.g., dual-purpose crops, high-yielding crops) [1.1.1] | FS | AD | |
| | | Modern biotechnology and genetic engineering (e.g., genetically modified stress-tolerant crops) [1.1.2] | FS | AD | |
| | Improved crop management [1.2] | Conservation agriculture [1.2.1] | | AD | |
| | | Integrated pest and weed management [1.2.2] | | AD | |
| | | Landscape pollination management [1.2.3] | | AD | |
| | | Organic agriculture [1.2.4] | | AD | |
| | Crop residue management [1.3] | No-till/minimum tillage; cover cropping; mulching [1.3.1] | FS | AD | |
| Soil [2] | Nutrient management [2.1] | Composting; appropriate fertilizer and manure use; precision farming [2.1.1] | FS | | |
| | Soil management [2.2] | Crop rotations, fallowing (green manures), intercropping with leguminous plants, conservation tillage [2.2.1] | | AD | |
| Water [3] | Water use efficiency and management [3.1] | Supplemental irrigation/water harvesting [3.1.1] | FS | AD | |
| | | Irrigation techniques to maximize water use (amount, timing, technology) [3.1.2] | FS | AD | |
| | | Modification of cropping calendar [3.1.3] | FS | AD | |
| Livestock [4] | Improved feed management [4.1] | Improving feed quality: diet supplementation; low-cost fodder conservation technologies [4.1.1] | FS | | MI |
| | | Improved grass species [4.1.2] | FS | | MI |
| | Altering integration within the system [4.2] | Alteration of animal species and breeds; the crop–livestock and crop–pasture ratios [4.2.1] | | AD | |
| | Livestock management [4.3] | Improved breeds and species (e.g., heat-tolerant breeds) [4.3.1] | | AD | MI |
| | | Infrastructure adaptation measures (e.g., housing, shade) [4.3.2] | | AD | |
| | | Animal disease and health [4.3.3] | FS | AD | |
| | Grazing management [4.4] | Adjust stocking densities to feed availability [4.4.1] | FS | | MI |
| | | Rotational grazing [4.4.2] | | AD | MI |
| | Manure management [4.5] | Anaerobic digesters for biogas and fertilizer [4.5.1] | FS | AD | MI |
| | | Composting; improved manure handling and storage (e.g., covering manure heaps); application techniques [4.5.2] | FS | | MI |
| **Total number of practices** | | | **FS: 14, AD: 20, MI: 7** | | |

According to Table 6, adaptation to climate change was the main target of CSA policies in Uganda, with 15 practices, accounting for 65% of adaptation practices. Food security was the second-most-targeted, with 10 practices accounting for 55% of food security practices, while only 5 mitigation practices were targeted, accounting for 57% of mitigation practices. Observations in [38] identified adaptation practices as the best option to reduce the negative impact of climate change on agriculture. The need to design efficient adaptation measures is even more critical, because agriculture is the main source of livelihood for about 80% of the rural population [19].

**Table 6.** Target practices and expected area of impact of climate-smart agriculture policies for Uganda.

| Sector | Description of Country-Level Policies | CSA Practice Target | Expected Impact |
|---|---|---|---|
| Crop | Promote highly adaptive and productive crop varieties and cultivars in drought-prone, flood-prone, and rainfed crop farming systems | [1.1.1] [1.1.2] | FS, AD FS, AD |
| Crop | Promote conservation agriculture and ecologically compatible cropping systems | [1.1.3] | AD |
| Water | Promote water harvesting and irrigation farming | [3.1.1] | FS, AD |
| Crop | Promote agricultural diversification and improved postharvest handling, storage, value addition, and marketing | [1.2.1] | AD |
| Livestock | Promote highly adaptive and productive livestock breeds | [4.3.1] | AD, MI |
| Livestock | Promote technologies for improved livestock feeds/feeding and sustainable management of rangelands and pastures through integrated rangeland management | [4.1.1] [4.1.2] [4.4.1] | FS, AD, MI FS, AD, MI FS, AD, MI |
| Livestock | Promote sustainable animal health management systems | [4.3.3] | FS, AD |
| Livestock | Encourage and promotion of dry season livestock feeding through pasture preservation and other feeding practices | [4.1.1] | FS, AD, MI |
| Livestock | Provide vaccination services for animal vector disease control, stock vaccines, and essential drugs for all notifiable diseases | [4.3.3] | FS, AD |
| Crop | Strengthen capacity for pest, weed, disease, and vermin control at all levels | [1.2.2] | AD |
| Water | Support development and sustainable use, management, and maintenance of water and land resources for agriculture | [3.1.1] [3.1.2] | FS, AD FS, AD |
| Total score | FS: 11 (7/14), AD: 15 (13/20), MI: 5 (4/7) | | |

**TOTAL AND WEIGHTED SCORE FOR EXPECTED CSA AREAS OF IMPACT IN UGANDA, KENYA AND MALAWI**

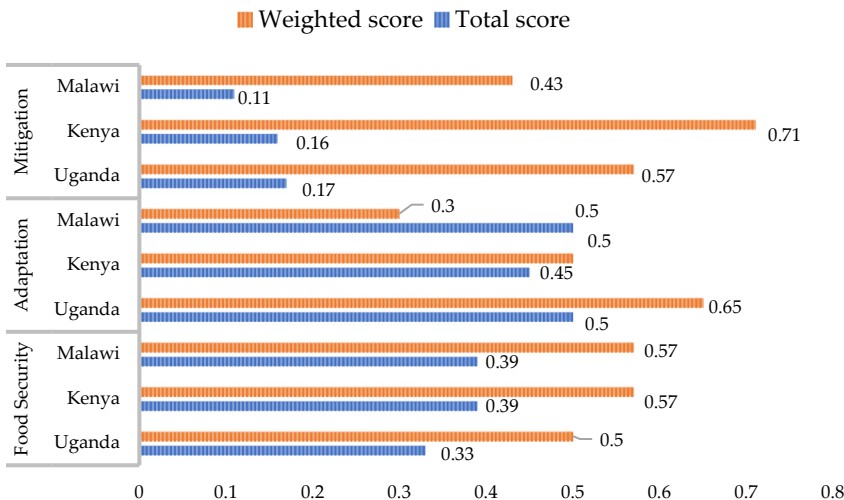

**Figure 10.** Scoring of expected CSA areas of impact in Uganda, Kenya, and Malawi.

**Table 7.** Target practices and expected area of impact of climate-smart agriculture policies for Kenya.

| Sector | Description of Country-Level Policies | CSA Practice Target | Expected Impact |
|---|---|---|---|
| Crop and Livestock | Promote crop varieties, livestock and fish breeds, and tree species that are adapted to varied weather conditions and tolerant to associated emerging pests and diseases | [1.1.1]<br>[1.1.2]<br>[4.3.1] | FS, AD<br>FS, AD<br>AD, MI |
| All | Promote sustainable management and utilization of natural resources | [1.2.1]<br>[2.1.1]<br>[3.1.1]<br>[3.1.2]<br>[4.4.1]<br>[4.4.2] | AD<br>FS<br>FS, AD<br>FS, AD<br>FS, AD, MI<br>AD, MI |
| Water | Promote water harvesting and storage, irrigation infrastructure development, and efficient water use | [3.1.1]<br>[3.1.2] | FS, AD<br>FS, AD |
| Crop and Livestock | Promote and support conservation and propagation of germplasm of species with adaptive capacity | [1.1.1]<br>[1.1.2]<br>[4.2.1] | FS, AD<br>FS, AD<br>AD |
| Livestock | Reduce the rate of emissions from livestock (manure and enteric fermentation) | [4.5.1]<br>[4.5.2] | FS, AD, MI<br>FS, MI |
| Total score | FS: 12 (8/14), AD: 14 (10/20), MI: 5 (5/7) | | |

**Table 8.** Target practices and expected area of impact of climate-smart agriculture policies for Malawi.

| Sector | Description of Country-Level Policies | CSA Practice Target | Expected Impact |
|---|---|---|---|
| Crop and Livestock | Facilitate access to high-quality farm inputs, including inorganic and organic fertilizer, improved seed and livestock breeds, and fish fingerlings | [1.1.1]<br>[1.1.2]<br>[1.2.4]<br>[4.2.1] | FS, AD<br>FS, AD<br>AD<br>AD |
| All | Promote investments in climate-smart agriculture and sustainable land and water management | [1.2.1]<br>[2.1.1]<br>[3.1.1]<br>[3.1.2]<br>[4.4.1]<br>[4.4.2] | AD<br>FS<br>FS, AD<br>FS, AD<br>FS, AD, MI<br>AD, MI |
| Crop and Livestock | Provide incentives to farmers to diversify their crop, livestock, and fisheries production and utilization | [1.1.1]<br>[1.1.2]<br>[4.3.1] | FS, AD<br>FS, AD<br>AD, MI |
| Water | Promote efficient and sustainable use of water in all irrigation schemes | [3.1.1]<br>[3.1.2] | FS, AD<br>FS, AD |
| Total score | FS: 11 (8/14), AD: 14 (6/20), MI: 3 (3/7) | | |

In Kenya, adaptation (14 measures) was the main target of CSA policies, although only 50% of adaptation practices were targeted, as presented in Table 7. This is motivated by the large proportion of the rural population depending on agriculture as a source of livelihood [16]. On the other hand, there were fewer practices targeting food security (12 practices), but more diversity of FS practices was mentioned (57%). More than 71% of

mitigation practices were targeted in the Kenya CSA policies (Table 7). According to the Kenya Climate Smart Agriculture Strategy 2017–2026 [39], the high emphasis on mitigation measures for Kenya could have been motivated by the high contribution of the agricultural sector to greenhouse gas emissions in the country, estimated at 67% of total GHG emissions.

Climate-smart agriculture policies for Malawi also targeted more adaptation practices (14) than food security (11) and mitigation (3) practices, as shown in Table 8. Like in Kenya and Uganda, this could be motivated by the fact that most of the rural population rely on agriculture as a source of livelihood [16]. However, policies in Malawi targeted more distinct food security (57%) and mitigation (43%) practices than adaptation (30%) practices, as shown in Table 8. The pre-eminence of food security in Malawi's CSA policies seems to be motivated by the government goal to achieve increased productivity for crops and livestock as well as agricultural GDP in the country. Malawi continues to implement the nationwide agricultural input subsidy program that targets smallholder farmers with a minimum amount of hybrid maize or legume seed and livestock plus one 50 Kg bag of both basal and top-dressing fertiliser. The observation that such use of inorganic fertiliser can have a negative impact on soil health [12,40] could be the reason why policies in Malawi are promoting mitigation for increased sustainable production to achieve food security [41].

According to Figure 10, the overall assessment indicates that Kenya had the highest weighted score for policies targeting mitigation of climate change (0.71), while Uganda scored highest for adaptation to climate change (0.65). Kenya and Malawi both had the highest score (0.57) for food security.

## 6. Conclusions

We assessed the intertemporal and spatial changes in anthropogenic (human-related) decisions to allocate land to a basket of CSA technologies in SSA. We also analysed the influence of the policy context in promoting the scaling of CSA practices. Our data were from Malawi, Kenya, and Uganda. We used both the quantitative national household-level data and qualitative policy-related data to respond to our objectives. Overall trends of land use in the study area show that in the past decade, agriculture and forest land have been increasing and decreasing, respectively. At the same time, the area under irrigation has been increasing, although at a low rate with regards to the total area under irrigation against the total agricultural area across the countries. At the household level, the average number of households allocating land and the extent of land allocation to a basket of CSA technologies has been fluctuating, but with an overall negative trend in the recent past. Our results show that the increase in owned land is positively associated with allocating more land to CSA. However, for rented-in land, increasing land is also associated with increased use of CSA technologies where land scarcity is high, but where land rental markets are not active, rented land indicated a negative association, although this was not significant.

Regarding adaptation, a one-year-lagged rainfall amount is associated with a decrease in land allocated to CSA in the subsequent year. A shock, such as a downside or upside rainfall deviation (associated with drought or flood), is likely to increase the number of households using CSA technologies but not the extent of agricultural land allocated to CSA. However, these results are context-specific. The differences in the policy emphasis on either adaptation, mitigation, or food security further show how different countries are promoting CSA technologies. Our analysis shows that in Kenya, CSA-related policies emphasize mitigation and food security, while in Uganda, the emphasis is on adaptation, and in Malawi, the emphasis is on food security.

Therefore, apart from promoting utilization of the different CSA technologies, agricultural policies in Sub-Saharan Africa (SSA) should also focus on access to land and the extent to which land allocation decisions are influencing the household decisions to allocate agricultural land to CSA technologies and practices across space and over time. This should promote resource-efficient production, especially among smallholder farmers. In addition to assessing access to land, future studies can assess differentiated impacts of land ownership and CSA adoption decision. For instance, this can include various characteristics such

as gender, experience, education, crops, and ecological zones. Regarding adaptation, future studies can also look at temperature variations in addition to rainfall variations.

**Author Contributions:** Conceptualization, S.E.T., D.N., S.P.K., and O.K.; methodology, S.E.T., D.N., S.P.K., and O.K.; software, S.E.T.; validation, S.E.T., D.N., S.P.K., and O.K.; formal analysis, S.E.T. and O.K.; investigation, S.E.T., D.N., G.N., S.P.K., and O.K.; resources, S.E.T., D.N., S.P.K., and O.K.; data curation, S.E.T. and O.K.; writing—original draft preparation, S.E.T., D.N., and O.K.; writing—review and editing, S.E.T. and O.K.; investigation, S.E.T., D.N., G.N., S.P.K., and O.K.; visualization, S.E.T. and O.K.; investigation, S.E.T., D.N., G.N., and O.K.; supervision, S.E.T. and O.K.; investigation, S.E.T., D.N., G.N., S.P.K., and O.K.; project administration, S.E.T., D.N., and O.K.; funding acquisition, S.E.T., D.N., and O.K. All authors have read and agreed to the published version of the manuscript.

**Funding:** This research was funded by the African Economic Research Consortium under the Climate Change and Economic Development in Africa (CCEDA) collaborative research project grant number RC21595 and the APC was also funded by the African Economic Research Consortium. The findings, opinions and recommendations are those of the authors and do not necessarily reflect the views of the Consortium, its individual members or the AERC Secretariat.

**Institutional Review Board Statement:** Not Applicable.

**Informed Consent Statement:** We used secondary data, and the National Statistics Offices in the study countries obtained informed consent from all subjects involved in the study. The public data are anonymized.

**Data Availability Statement:** We used the Living Standards Measurement Surveys (LSMS) collected by National Statistical Offices and the World Bank. The data are publicly available on https://microdata.worldbank.org/index.php/catalog/lsms/?page=1&ps=15&repo=lsms (accessed on 5 October 2021).

**Acknowledgments:** We acknowledge the Ministry of Agriculture, Malawi; Makerere University, Uganda; Lilongwe University of Agriculture and Natural Resources (LUANAR), Malawi; Leibniz Institute of Agricultural Engineering and Bioeconomy, Postdam, Germany; and Egerton University, Kenya, for creating a conducive environment for us to work on this paper. We also appreciate the technical support from African Economic Research Consortium and fellow researchers under the programme Grants for Collaborative Research on Climate Change for their input.

**Conflicts of Interest:** We declare no conflict of interest. The funders had no role in the design of the study; in the collection, analyses, or interpretation of data; in the writing of the manuscript; or in the decision to publish the results.

## Appendix A

**Table A1.** Estimated coefficients from Probit and Tobit Models for CSA dummy and Total area allocated to CSA in Malawi.

| Variables | Estimated Coefficients in Malawi | | | | | | | |
|---|---|---|---|---|---|---|---|---|
| | Parsimonious Models | | Models with Control Variables | | Parsimonious Models | | Models with Control Variables | |
| | Probit1 | Probit2 | Probit3 | Probit4 | Tobit1 | Tobit2 | Tobit3 | Tobit4 |
| **Key Variables** | | | | | | | | |
| Rent-in dummy | 0.63 **** | 0.63 **** | 0.71 **** | 0.73 **** | 0.30 **** | 0.29 **** | 0.33 **** | 0.33 **** |
| | (0.09) | (0.09) | (0.09) | (0.09) | (0.04) | (0.04) | (0.04) | (0.04) |
| One-year-lag Rainfall (per 100 mm) | −0.05 *** | | −0.05 *** | | −0.03 *** | | −0.04 **** | |
| | (0.02) | | (0.02) | | (0.01) | | (0.01) | |
| Upside rainfall deviation (dm) | | 0.00 | | 0.01 ** | | −0.00 | | 0.00 |
| | | (0.00) | | (0.00) | | (0.00) | | (0.00) |
| Downside rainfall deviation (dm) | | 0.00 | | 0.02 **** | | −0.00 | | 0.00 |
| | | (0.00) | | (0.00) | | (0.00) | | (0.00) |
| Owned land (GPS measured—ha) | | | 0.71 **** | 0.72 **** | | | 0.30 **** | 0.30 **** |
| | | | (0.09) | (0.09) | | | (0.01) | (0.01) |
| **Control Variables** | | | −0.08 | −0.10 | | | −0.07 ** | −0.08 ** |
| Sex of household head | | | (0.07) | (0.07) | | | (0.04) | (0.04) |
| | | | 0.00 | 0.00 | | | 0.00 **** | 0.00 **** |
| Age of household head | | | (0.00) | (0.00) | | | (0.00) | (0.00) |
| | | | −0.01 * | −0.01 * | | | −0.01 ** | −0.01 ** |
| Education of household head | | | (0.01) | (0.01) | | | (0.00) | (0.00) |
| | | | −0.29 ** | −0.30 ** | | | −0.10 | −0.10 |
| Share of male labour out of total labour | | | (0.14) | (0.14) | | | (0.07) | (0.07) |
| | | | 0.13 * | 0.12 * | | | 0.06 **** | 0.07 **** |
| Household size | | | (0.07) | (0.07) | | | (0.02) | (0.02) |
| | | | 0.07 | 0.06 | | | 0.00 | 0.00 |
| TLU per labour ratio | | | (0.05) | (0.05) | | | (0.00) | (0.00) |
| | | | 0.01 **** | 0.01 **** | | | 0.01 **** | 0.01 **** |
| One-year-lag TLU per labour ratio | | | (0.00) | (0.00) | | | (0.00) | (0.00) |
| | | | 0.01 | 0.01 | | | 0.03 **** | 0.02 **** |
| Distance to urban centre | | | (0.01) | (0.01) | | | (0.01) | (0.01) |
| | | | −0.03 | −0.03 | | | −0.04 *** | −0.04 *** |
| Household size per labour ratio | | | (0.03) | (0.03) | | | (0.02) | (0.02) |

**Table A1.** *Cont.*

| Variables | Estimated Coefficients in Malawi | | | | | | | |
|---|---|---|---|---|---|---|---|---|
| | Parsimonious Models | | Models with Control Variables | | Parsimonious Models | | Models with Control Variables | |
| | Probit1 | Probit2 | Probit3 | Probit4 | Tobit1 | Tobit2 | Tobit3 | Tobit4 |
| Capital asset index | | | −0.27 **** | −0.28 **** | | | −0.12 **** | −0.12 **** |
| | | | (0.04) | (0.04) | | | (0.02) | (0.02) |
| | | | (0.04) | (0.04) | | | (0.02) | (0.02) |
| Base Year (2013) | | | | | | | | |
| 2016 panel year | 0.32 **** | 0.23 **** | 0.30 **** | 0.22 **** | 0.14 **** | 0.08 *** | 0.13 **** | 0.08 ** |
| | (0.06) | (0.06) | (0.06) | (0.06) | (0.03) | (0.03) | (0.03) | (0.03) |
| 2019 panel year | 0.16 *** | 0.15 *** | 0.25 **** | 0.23 **** | 0.01 | 0.01 | 0.08 *** | 0.07 ** |
| | (0.05) | (0.05) | (0.06) | (0.06) | (0.03) | (0.03) | (0.03) | (0.03) |
| Constant | 0.16 | −0.26 **** | −0.22 | −0.71 **** | 0.09 | −0.10 *** | −0.26 ** | −0.56 **** |
| | (0.15) | (0.06) | (0.21) | (0.17) | (0.08) | (0.03) | (0.10) | (0.08) |
| lnsig2u | −0.25 ** | −0.24 ** | −0.99 **** | −1.02 **** | | | | |
| | (0.10) | (0.10) | (0.15) | (0.16) | | | | |
| sigma_u | | | | | 0.54 **** | 0.54 **** | 0.32 **** | 0.33 **** |
| | | | | | (0.02) | (0.02) | (0.02) | (0.02) |
| sigma_e | | | | | 0.62 **** | 0.62 **** | 0.60 **** | 0.60 **** |
| | | | | | (0.01) | (0.01) | (0.01) | (0.01) |
| Panel households | 1439 | 1439 | 1439 | 1439 | 1439 | 1439 | 1439 | 1439 |
| Left censored (_n) | | | | | 2193 | 2193 | 2193 | 2193 |
| Uncensored (_n) | | | | | 2124 | 2124 | 2124 | 2124 |
| Observations | 4317 | 4317 | 4317 | 4317 | 4317 | 4317 | 4317 | 4317 |

Robust standard errors in parentheses **** $p < 0.001$, *** $p < 0.01$, ** $p < 0.05$, * $p < 0.1$.

**Table A2.** Estimated coefficients from Probit and Tobit Models for CSA dummy and Total area allocated to CSA in Uganda.

| Variables | Estimated Coefficients in Uganda | | | | | | | |
|---|---|---|---|---|---|---|---|---|
| | **Parsimonious Models** | | **Models with Control Variables** | | **Parsimonious Models** | | **Models with Control Variables** | |
| | **Probit1** | **Probit2** | **Probit3** | **Probit4** | **Tobit1** | **Tobit2** | **Tobit3** | **Tobit4** |
| Key Variables | | | | | | | | |
| Rent-in dummy (1 = Yes) | −0.07 | −0.07 | −0.09 | −0.09 | −0.08 | −0.09 | −0.11 | −0.12 |
| | (0.15) | (0.15) | (0.15) | (0.15) | (0.20) | (0.20) | (0.19) | (0.20) |
| Irregular rains (1 = Yes) | 0.01 | | 0.02 | | −0.01 | | −0.00 | |
| | (0.18) | | (0.18) | | (0.22) | | (0.22) | |
| Drought (1 = Yes) | | −0.10 | | −0.06 | | −0.14 | | −0.10 |
| | | (0.14) | | (0.14) | | (0.17) | | (0.17) |
| Floods (1 = Yes) | | 0.10 | | 0.16 | | 0.06 | | 0.14 |
| | | (0.20) | | (0.20) | | (0.26) | | (0.25) |
| Owned land (Self-reported ha) | | | 0.02 | 0.02 | | | 0.04 | 0.04 |
| | | | (0.03) | (0.03) | | | (0.05) | (0.04) |
| Control Variables | | | | | | | | |
| Sex of household head (1 = Female) | | | 0.01 | 0.01 | | | −0.01 | −0.02 |
| | | | (0.16) | (0.17) | | | (0.21) | (0.21) |
| Age of household head | | | 0.00 | −0.00 | | | −0.00 | −0.00 |
| | | | (0.00) | (0.00) | | | (0.01) | (0.01) |
| Education of household head | | | −0.02 | −0.02 | | | −0.02 | −0.02 |
| | | | (0.02) | (0.02) | | | (0.02) | (0.02) |
| Share of male to total labour | | | −0.32 | −0.33 | | | −0.33 | −0.34 |
| | | | (0.33) | (0.34) | | | (0.43) | (0.43) |
| Total livestock unit to labour ratio | | | −0.27 ** | −0.27 ** | | | −0.35 ** | −0.35 ** |
| | | | (0.12) | (0.12) | | | (0.16) | (0.16) |
| One-year-lag TLU to labour ratio | | | 0.17 | 0.18 | | | 0.20 | 0.20 |
| | | | (0.15) | (0.15) | | | (0.21) | (0.21) |
| Urban (1 = Rural) | | | 0.63 ** | 0.64 ** | | | 0.77 * | 0.79 ** |
| | | | (0.30) | (0.29) | | | (0.40) | (0.40) |
| Household-to-labour ratio | | | −0.09 | −0.09 | | | −0.06 | −0.06 |
| | | | (0.10) | (0.09) | | | (0.17) | (0.17) |
| Capital asset index | | | 0.25 *** | 0.25 *** | | | 0.33 *** | 0.33 *** |
| | | | (0.09) | (0.10) | | | (0.11) | (0.11) |
| Base Year (2019) | | | | | | | | |

**Table A2.** *Cont.*

| Variables | Estimated Coefficients in Uganda | | | | | | | |
|---|---|---|---|---|---|---|---|---|
| | **Parsimonious Models** | | **Models with Control Variables** | | **Parsimonious Models** | | **Models with Control Variables** | |
| | **Probit1** | **Probit2** | **Probit3** | **Probit4** | **Tobit1** | **Tobit2** | **Tobit3** | **Tobit4** |
| 2020 panel year | 1.10 **** | 1.08 **** | 1.25 **** | 1.24 **** | 1.33 **** | 1.31 **** | 1.46 **** | 1.43 **** |
| | (0.19) | (0.19) | (0.24) | (0.24) | (0.22) | (0.22) | (0.24) | (0.24) |
| Constant | −1.97 **** | −1.93 **** | −2.19 **** | −2.14 **** | −2.51 **** | −2.44 **** | −2.72 **** | −2.65 **** |
| | (0.24) | (0.25) | (0.56) | (0.55) | (0.30) | (0.31) | (0.73) | (0.73) |
| lnsig2u | −4.18 | −3.71 | −11.34 | −13.90 | | | | |
| | (12.87) | (8.31) | (17,807.10) | (232,114.49) | | | | |
| sigma_u | | | | | 0.49 * | 0.50 * | 0.33 | 0.34 |
| | | | | | (0.27) | (0.27) | (0.39) | (0.37) |
| sigma_e | | | | | 1.22 **** | 1.22 **** | 1.22 **** | 1.22 **** |
| | | | | | (0.16) | (0.16) | (0.15) | (0.15) |
| Number of hhid_2019 | 407 | 407 | 407 | 407 | 407 | 407 | 407 | 407 |
| Left censored (_n) | | | | | 727 | 727 | 727 | 727 |
| Uncensored (_n) | | | | | 87 | 87 | 87 | 87 |
| Observations | 814 | 814 | 814 | 814 | 814 | 814 | 814 | 814 |

Robust standard errors in parentheses **** $p < 0.001$, *** $p < 0.01$, ** $p < 0.05$, * $p < 0.1$.

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
