# Peer review of "Anthropogenic Land Use Change and Adoption of Climate Smart Agriculture in Sub-Saharan Africa"

_sustainability, doi:10.3390/su142214729_

Round 1
Reviewer 1 Report
General comments:
It is an interesting study to investigate the low uptake of climate smart agriculture in Sub-Saharan Africa. The overall research significance, used methods, and presentation is good. The used language is also simple and understandable. I have provided few suggestions to further improve the presentation of this manuscript
Abstract:
· Abstract has nicely covered all the required sections. However, the provided recommendation on access to land is very generic. Access to land and water and other technologies are understandably important for upscaling the CSA, but policy makers search for more specific recommendation. I suggest to make it specific based on the study results or literature review.
Introduction:
· Beside, conducting this research in the adopted study area, the study framework is another main innovation of this research. However, the established hypotheses are weakly supported by the literature review. Please extend your literature review discussion on establishing three hypothesis.
Methodology:
· Please add the study area map
· As various datasets are used in this research, add a table describing the data unit, period, and acquired sources.
· Schematic flow diagram is always helpful in understanding the research methodology, I suggest to add it here.
· A brief literature discussion on controlled variables in the methodology or introduction section is also needed
Results and discussion:
· I read the results section and got the sense that tables and graphs presentation can be improved to avoid confusion and keep the interest of readers alive. Please keep the sequence of tables and figures and the related text descriptions. For instance, two tables are numbered as table 1.
· One major concern about the regression is that the study uses many similar independent variable such as flood and rainfall. I wonder if the authors tested the collinearity amongst them before selecting for regression?
Author Response
REVIEWER 1
General comments:
It is an interesting study to investigate the low uptake of climate smart agriculture in Sub-Saharan Africa. The overall research significance, used methods, and presentation is good. The used language is also simple and understandable. I have provided few suggestions to further improve the presentation of this manuscript
Overall Response
- We appreciate the time taken to review our paper and the favourable comments. We have revised the paper accordingly.
Abstract:
Comment 1
- Abstract has nicely covered all the required sections. However, the provided recommendation on access to land is very generic. Access to land and water and other technologies are understandably important for upscaling the CSA, but policy makers search for more specific recommendation. I suggest to make it specific based on the study results or literature review.
Response 1
- We have revised the abstract to end with the statement below
- “Therefore, scaling up CSA in SSA will require that agricultural-related policies promote land tenure security and land markets while promoting climate smart farming for food security, adaptation and mitigation.”
Introduction:
Comment 2
- Beside, conducting this research in the adopted study area, the study framework is another main innovation of this research. However, the established hypotheses are weakly supported by the literature review. Please extend your literature review discussion on establishing three hypothesis.
Response 2
We have added a paragraph based on the literature review
- Several factors, including household, community, and governance factors, influence household decisions, for instant land allocated to CSA [24]. A review of the literature shows that household decisions are a function of household characteristics (gender, size, location, access to community services, education, and experience, and household endowments like land and asset wealth) [24-26]. Additionally, climatic factors like rainfall patterns and governance factors like policies and legislation can influence production decisions over time [24-27]. If a household experience a deviation in rainfall pattern like drought and flood in one year, their production decisions are likely to change in the sub-sequent year to minimize loss or risk [13, 23]. Figure 1 conceptualizes the discussed household decision to allocate land for CSA technologies and the factors likely to affect that decision, as discussed.
Reference
- Ariom, T. O.; Dimon, E.; Nambeye, E.; Diouf, N. S.; Adelusi, O. O.; Boudalia, S. Climate-Smart Agriculture in African Countries: A Review of Strategies and Impacts on Smallholder Farmers. S 2022. 14(18), 11370.
- Opeyemi, G.; Opaluwa, H.I.; Adeleke, A.O.; Ugbaje, B. Effect of climate smart agricultural practices on farming householdsʹ food security status in Ika North East Local Government Area, Delta State, Nigeria. AFSci 2021.19(2), 30-42.
- Bazzana, D.; Foltz, J.; Zhang, Y. Impact of climate smart agriculture on food security: an agent-based analysis. FP 111, 102304.
- Abegunde, V.O.; Sibanda, M.; Obi, A. Effect of climate-smart agriculture on household food security in small-scale production systems: A micro-level analysis from South Africa. CSS 8(1), 2086343.
Methodology:
Comment 3
- Please add the study area map
Response 3
- We have added the Map of Africa showing Malawi, Kenya and Uganda as the study area in the introduction.
Comment 4
- As various datasets are used in this research, add a table describing the data unit, period, and acquired sources.
Response 4
- We have added the table below in the data section.
Data and Data Source
Country |
Data type |
Period |
Source |
Objective |
Malawi |
FAOSTAT |
1961 - 2008 |
FAO https://www.fao.org/statistics/en/ |
1 |
|
LSMS |
2013, 2016/17, 2019/20 |
Word Bank - LSMS https://www.worldbank.org/en/programs/lsms |
2 and 3 |
|
Secondary data |
Overtime |
Literature review |
4 |
|
|
|
|
|
Uganda |
FAOSTAT |
1961 - 2008 |
FAO https://www.fao.org/statistics/en/ |
1 |
|
LSMS |
2014/15, 2019/20 |
Word Bank - LSMS https://www.worldbank.org/en/programs/lsms |
2 and 3 |
|
Secondary data |
Overtime |
Literature review |
4 |
|
|
|
|
|
Kenya |
FAOSTAT |
1961 - 2008 |
FAO https://www.fao.org/statistics/en/ |
1 |
|
Secondary data |
Overtime |
Literature review |
4 |
Comment 5
- Schematic flow diagram is always helpful in understanding the research methodology, I suggest to add it here.
Response 5
- We have developed the flow diagram below. This is included in the theoretical section of the paper.
Figure 2: Flow diagram of key research variables and objectives
Note: Solid line arrows show a direct pathway in a year while the dashed arrows show long-term pathways overtime.
Comment 6
- A brief literature discussion on controlled variables in the methodology or introduction section is also needed.
Response 6
- This is covered in the revised introduction section as presented above
Results and discussion:
Comment 7
- I read the results section and got the sense that tables and graphs presentation can be improved to avoid confusion and keep the interest of readers alive. Please keep the sequence of tables and figures and the related text descriptions. For instance, two tables are numbered as table 1.
Response 7
- We have ordered the tables and figures in chronological order and revised the text across the different sections to reflect the list.
Comment 8
- One major concern about the regression is that the study uses many similar independent variable such as flood and rainfall. I wonder if the authors tested the collinearity amongst them before selecting for regression?
Response 8
- The analysis carefully separated the rainfall variables that are related across the models (Probit and Tobit models 1 to 4) because of data availability. For instance, drought and flood dummy variables are used in Uganda models while upward and downward one-year lagged rainfall deviations are used in the Malawi models. Again, the analysis in Malawi has separately analysed overall one-year lagged rainfall variable from upward and downward one-year lagged rainfall deviations. The variables in Malawi are generated from the same data hence highly correlated. Since the variables are used separately, we do not anticipate a multi-collinearity problem in the analysis.

Reviewer 2 Report
REVIEW FORM
Sustainability-1937264
Anthropogenic Land Use Change and Adoption of Climate Smart Agriculture in Sub-Saharan Africa
Comments from the reviewer
1. Abstract
- Some keywords should be added like “Sub-Saharan Africa”, “Climate Smart Agriculture”.
2. Introduction
- The introduction part is found to be too long.
- The reference should be provided for the sentence “Livestock production alone contributes about 14.5 percent of global GHG emissions and nearly half of the agriculture sector’s emissions, from enteric fermentation and land clearing (i.e. animal digestion, feed production, manure management, and forest cover loss). (in the first four lines of page 2)”
- Please be consistent with the way of writing of the term “Climate-Smart Agriculture” or “Climate Smart Agriculture”.
- Sub-sections “1.1 Climate-Smart Agriculture and SSA in Context” and “1.2 Theoretical framework” should be moved to the other section.
3. Materials and Methods
- In sub-section “2.1 Data Sources”, please provide the years for the “Living Standards Measurement Surveys (LSMS)” and “FAOSTAT data (in lines 183 of page 4)”.
4. Results
- Please provide the year for the “FAOSTAT data (in lines 334 of page 7 and lines 385 of page 9)”.
- In sub-section “3.2 Empirical Results and Discussion”, please check for the abbreviation “SCA (in lines 376 of page 8)”.
- In lines 379 – 381, the authors stated that “In Uganda, the trend of households using CSA shows a big drop in the 2019 panel year, which was only 3 percent compared to 14 and 13 percent in the 2015 and 2020 survey rounds.”, is it correct for the figure of 3% and should it be 4%?
- In lines 400 – 402, please refer to Figure 5 for the sentence “From Malawi, the results indicate that the most used technologies are soil control bunds, vetiver grass, organic manure and the least used technologies relate to small-scale irrigation.”.
- In lines 443 – 446, for the sentence “On the extent of land allocation (Tobit 4), a one-hectare increase in owned land increases the household land allocated to CSA technologies by 0.11 ha, which is significant at a 1 percent level.”, is it correct for the figure 1% or it should be 0.1% instead?
- In lines 458 – 460, for the sentence “From the probit model (probit 4), renting-in more land is associated with increased allocation of land to a basket of CSA technologies by 22 percent while increasing the extent by 0.12 ha, which is significant at 1 percent.”, please identify where the figure 0.12 ha can be found in Table 1.
- In lines 506 – 508, for the sentence “If one-year lagged rainfall increases by 100mm, households are likely to reduce the land allocated to CSA by 1 percentage point while reducing the area by 0.01ha.”, please explain clearly how can the numbers indicated in the above sentence (like 1 percentage point and 0.01ha) link to Table 1.
- In lines 510 – 512, check the sentence “Assessing the upside and downside deviation variables, probit model 3 presents a positive and significant effect of both downside rainfall deviation (more associated with drought) and upside rainfall deviation (mores associated with floods).”, whether it is correct or not?
- In lines 521 – 523, please indicate the numbers appeared in the sentence “The CSA profile for Uganda indicates that 25 percent of cropped land is under root crops while 17 percent is under banana production and cereals (maize, sorghum, millet, rice) take up 32 percent.”, how can it be found in Table 2?
- In lines 527 – 530, please elaborate more for the sentence “Using the Total Livestock Units to household labour ratio, we note that more livestock compared to household labour is associated with reduced agricultural land area allocated to CSA technologies as expected.”.
- In line 542, the authors refer to Figure 5, is it correct? Should it be Figure 8?, as well as in line 574.
- In lines 544 – 545, focusing at the sentence “From Table 4, adaptation to climate change was the main target of CSA policies in Uganda with 15 practices integrating 60% of adaptation practices.”, should it be 65% of adaptation practices?
- In lines 545 – 547, focusing at the sentence “Food security was the second most targeted with 11 practices for 57% of food security practices, while mitigation only 6 practices for 57% of mitigation practices.”, should it be “10 practices for 50% of food security practices”, and “only 5 practices for 57% of mitigation practices”?
- In line 551, please make it clear for the number “14” indicated in the sentence “In Kenya (14),….”
- In lines 556 – 557, please add the reference for the “Kenya Climate-Smart Agriculture Strategy 2017-2026”
- Focusing on Table 4, at the total score of FS, should it be 11 instead of 10?
- Focusing on Tables 4 – 6, at the total scores of FS, AD, and MI, what is the numbers in the brackets with slashes? Please explain them clearly.
- In lines 610 – 612, at the sentence “Our analysis shows that in Kenya, CSA-related policies emphasize mitigation and in Uganda, the policy emphasis is on adaptation while for Malawi it is food security.”, should it be for “Malawi and Kenya”?
5. Conclusions
- Please change the order of section “Conclusions” from section 3. to be section “4. Conclusions”
6. General comments
- There are some typing errors in some places (e.g. raid-fed in lines 9 of section “1. Introduction”), please check them throughout the manuscript and correct them accordingly.
- Please use a proper way of writing the citation like in the 2nd paragraph of page 3 “…..irrigation [12, 13, 9].”. Please check and correct it throughout the manuscript.

Author Response
REVIEWER 2: REVIEW FORM
Sustainability-1937264: Anthropogenic Land Use Change and Adoption of Climate Smart Agriculture in Sub-Saharan Africa
Comments from the reviewer
- Abstract
Comment 1
- Some keywords should be added like “Sub-Saharan Africa”, and “Climate Smart Agriculture”.
Response 1
- We have included the suggested words and removed the specific country names since these are covered in the “Sub-Saharan Africa” word.
- Introduction
Comment 2
- The introduction part is found to be too long.
Response 2
- We have elevated the sub-sections in the introduction to sections. We have re-numbered all the sections in the revised paper.
Comment 3
- The reference should be provided for the sentence “Livestock production alone contributes about 14.5 percent of global GHG emissions and nearly half of the agriculture sector’s emissions, from enteric fermentation and land clearing (i.e. animal digestion, feed production, manure management, and forest cover loss). (in the first four lines of page 2)”
Response 3
- The reference document is the IPCC report following the preceding statements. We have cited the report accordingly.
Comment 4
- Please be consistent with the way of writing of the term “Climate-Smart Agriculture” or “Climate Smart Agriculture”.
Response 4
- We have maintained “Climate Smart Agriculture” in the paper.
Comment 5
- Sub-sections “1.1 Climate-Smart Agriculture and SSA in Context” and “1.2 Theoretical framework” should be moved to the other section.
Response 5
- We have revised the numbering of the sections and sub-sections.
- Materials and Methods
Comment 6
- In sub-section “2.1 Data Sources”, please provide the years for the “Living Standards Measurement Surveys (LSMS)” and “FAOSTAT data (in lines 183 of page 4)”.
Response 6
- We have added the following sentences
- The three rounds of data in Malawi were collected in 2013, 2016/17 and 2019/20. In Uganda, the two rounds of LSMS data used were collected in 2014/15 and 2019/20. We use the FAOSTAT data from 1961 to 2018 to assess country-level indicators, more specifically to respond to objective 1 for Kenya, where we could not access the national wide household survey data.
- Results
Comment 7
- Please provide the year for the “FAOSTAT data (in lines 334 of page 7 and lines 385 of page 9)”.
Response 7
- We have indicated (1990-2018) under Figure 1 and (1961-2018) under Figure 2
Comment 8
- In sub-section “3.2 Empirical Results and Discussion”, please check for the abbreviation “SCA (in lines 376 of page 8)”.
Response 8
- We have corrected and changed to CSA
Comment 9
- In lines 379 – 381, the authors stated that “In Uganda, the trend of households using CSA shows a big drop in the 2019 panel year, which was only 3 percent compared to 14 and 13 percent in the 2015 and 2020 survey rounds.”, is it correct for the figure of 3% and should it be 4%?
Response 8
- We have corrected the statistics. Indeed, it should be 4%.
Comment 9
- In lines 400 – 402, please refer to Figure 5 for the sentence “From Malawi, the results indicate that the most used technologies are soil control bunds, vetiver grass, organic manure and the least used technologies relate to small-scale irrigation.”.
Response 9
- We have properly referred to the Figure since it is no longer Figure 5 after changes in the earlier sections.
Comment 10
- In lines 443 – 446, for the sentence “On the extent of land allocation (Tobit 4), a one-hectare increase in owned land increases the household land allocated to CSA technologies by 0.11 ha, which is significant at a 1 percent level.”, is it correct for the figure 1% or it should be 0.1% instead?
Response 10
- The 1 percent level is for significance and not the magnitude of effect in the model. The Tobit model shows absolute magnitude hence the use of 0.11 ha and not 1.1% effect.
Comment 11
- In lines 458 – 460, for the sentence “From the probit model (probit 4), renting-in more land is associated with increased allocation of land to a basket of CSA technologies by 22 percent while increasing the extent by 0.12 ha, which is significant at 1 percent.”, please identify where the figure 0.12 ha can be found in Table 1.
Response 11
- We have added “…. as indicated in the Tobit model (Tobit 4)”
Comment 12
- In lines 506 – 508, for the sentence “If one-year lagged rainfall increases by 100mm, households are likely to reduce the land allocated to CSA by 1 percentage point while reducing the area by 0.01ha.”, please explain clearly how can the numbers indicated in the above sentence (like 1 percentage point and 0.01ha) link to Table 1.
Response 13
- We have revised the sentence to read
- From Table 1, if one-year lagged rainfall increases by 100mm, households are likely to reduce the land allocated to CSA by 1 percentage point as shown in probit 3 results while reducing the area under CSA by 0.01ha as assessed in Tobit 3 model results.
Comment 14
- In lines 510 – 512, check the sentence “Assessing the upside and downside deviation variables, probit model 3 presents a positive and significant effect of both downside rainfall deviation (more associated with drought) and upside rainfall deviation (mores associated with floods).”, whether it is correct or not?
Response 14
- The statement is based on observing 0.00** and 0.00**** for downside and upside deviations. We don’t explain the magnitude because the figures are two small )almost zero.
Comment 15
- In lines 521 – 523, please indicate the numbers appeared in the sentence “The CSA profile for Uganda indicates that 25 percent of cropped land is under root crops while 17 percent is under banana production and cereals (maize, sorghum, millet, rice) take up 32 percent.”, how can it be found in Table 2?
Response 15
- The statement is from literature and we have revised it to continue with the statement on livestock since the cited paper is the same. The revised statement reads
- The CSA profile for Uganda indicates that 25 percent of cropped land is under root crops while 17 percent is under banana production, cereals (maize, sorghum, millet, rice) take up 32 percent and livestock is key in the country [17].
Comment 16
- In lines 527 – 530, please elaborate more for the sentence “Using the Total Livestock Units to household labour ratio, we note that more livestock compared to household labour is associated with reduced agricultural land area allocated to CSA technologies as expected.”.
Response 16
- We have improved the following sentence to read
- The understanding is that an increase in livestock should increase the availability of manure. However, manure application can be labour intensive hence households with low labour are constrained in using manure in the field compared to using the manure for other uses like producing biogas if the system is established.
Comment 17
- In line 542, the authors refer to Figure 5, is it correct? Should it be Figure 8?, as well as in line 574.
Response 17
- Yes, it was to be Figure 8 but we have changed to Figure 9 because we added another figure earlier in the document.
Comment 17
- In lines 544 – 545, focusing at the sentence “From Table 4, adaptation to climate change was the main target of CSA policies in Uganda with 15 practices integrating 60% of adaptation practices.”, should it be 65% of adaptation practices?
Response 17
- Indeed, it should be 65%. We have changed.
Comment 18
- In lines 545 – 547, focusing at the sentence “Food security was the second most targeted with 11 practices for 57% of food security practices, while mitigation only 6 practices for 57% of mitigation practices.”, should it be “10 practices for 50% of food security practices”, and “only 5 practices for 57% of mitigation practices”?
Response 18
- We have changed to reflect the correct numbers
Comment 19
- In line 551, please make it clear for the number “14” indicated in the sentence “In Kenya (14),….”
Response 19
- The sentence was changed to “In Kenya, adaptation (14 measures) was the …”
Comment 20
- In lines 556 – 557, please add the reference for the “Kenya Climate-Smart Agriculture Strategy 2017-2026”
Response 20
- We have added to the reference list
Comment 21
- Focusing on Table 4, at the total score of FS, should it be 11 instead of 10?
Response 21
- We have changed the total score 11
Comment 22
- Focusing on Tables 4 – 6, at the total scores of FS, AD, and MI, what is the numbers in the brackets with slashes? Please explain them clearly.
Response 23
- We have added the following statement
- “From the tables, the numbers in brackets with slashes represent the order number of CSA practices by sector (integers), management objectives within the sectors (one decimal place) and practices within management objectives two decimal places).”
Comment 23
- In lines 610 – 612, at the sentence “Our analysis shows that in Kenya, CSA-related policies emphasize mitigation and in Uganda, the policy emphasis is on adaptation while for Malawi it is food security.”, should it be for “Malawi and Kenya”?
Response 23
- We have changed the sentences to read “Our analysis shows that in Kenya, CSA-related policies emphasize mitigation and food security, while in Uganda the emphasis is on adaptation and in Malawi the emphasis is on food security.”
- Conclusions
Comment 24
- Please change the order of section “Conclusions” from section 3. to be section “4. Conclusions”
Response 24
- With the changes made in the introduction, the conclusion section is now section 6
- General comments
Comment 25
- There are some typing errors in some places (e.g. raid-fed in lines 9 of section “1. Introduction”), please check them throughout the manuscript and correct them accordingly.
Response 25
- We have carefully looked at the errors and cleaned the paper, thank you for this observation.
Comment 26
- Please use a proper way of writing the citation like in the 2nd paragraph of page 3 “…..irrigation [12, 13, 9].”. Please check and correct it throughout the manuscript.
Response 26
- We have carefully revised the citations and references. Where we have more than one citation following each other, we have indicated it as a continuous citation. For instance, if the citation is from number 24 to 26, we have cited it as [24-26]. The other citations are given in chronological order.

Round 2
Reviewer 2 Report
The reviewer has no further comments as the authors followed and accommodated all the comments in the revised manuscript.